# Ventromedial medulla inhibitory neuron inactivation induces REM sleep without atonia and REM sleep behavior disorder

Sara Valencia Garcia [1,2], Frédéric Brischoux[1,2], Olivier Clément[1,2], Paul-Antoine Libourel[1,2], Sébastien Arthaud[1,2], Michael Lazarus [3], Pierre-Hervé Luppi [1,2] & Patrice Fort [1,2]

Despite decades of research, there is a persistent debate regarding the localization of GABA/glycine neurons responsible for hyperpolarizing somatic motoneurons during paradoxical (or REM) sleep (PS), resulting in the loss of muscle tone during this sleep state. Combining complementary neuroanatomical approaches in rats, we first show that these inhibitory neurons are localized within the ventromedial medulla (vmM) rather than within the spinal cord. We then demonstrate their functional role in PS expression through local injections of adeno-associated virus carrying specific short-hairpin RNA in order to chronically impair inhibitory neurotransmission from vmM. After such selective genetic inactivation, rats display PS without atonia associated with abnormal and violent motor activity, concomitant with a small reduction of daily PS quantity. These symptoms closely mimic human REM sleep behavior disorder (RBD), a prodromal parasomnia of synucleinopathies. Our findings demonstrate the crucial role of GABA/glycine inhibitory vmM neurons in muscle atonia during PS and highlight a candidate brain region that can be susceptible to α-synuclein-dependent degeneration in RBD patients.

---

[1] SLEEP Team, Neuroscience Research Center of Lyon - CRNL, CNRS UMR 5292, INSERM U1028, Lyon, France. [2] Lyon I - Claude Bernard University (UCBL), Lyon, France. [3] International Institute for Integrative Sleep Medicine, University of Tsukuba, Tsukuba, Japan. Correspondence and requests for materials should be addressed to P.F. (email: patrice.fort@univ-lyon1.fr)

Paradoxical sleep (PS), or rapid eye movement (REM) sleep, is characterized by a cortical activation associated with a generalized muscle atonia. REM sleep behavior disorder (RBD) is a parasomnia characterized by the loss of this paralysis, allowing patients to execute abnormal movements and dream enactments during PS[1,2]. Recent longitudinal studies revealed that ≈80% of patients suffering idiopathic RBD develop a synucleinopathy such as Parkinson's disease with a latency of 10–15 years since the onset of RBD symptoms[3–6]. Hence, disentangling neuronal networks responsible for muscle atonia during PS may help to understand RBD pathogenesis.

Somatic motoneurons are hyperpolarized specifically during PS by a barrage of high-amplitude inhibitory post-synaptic potentials with glycinergic neurotransmission playing an essential role in this inhibitory process[7–10]. A synergistic contribution of GABA has been reported[11]. Although it is currently assumed that GABA/glycine pre-motoneurons activated specifically during PS underlie muscle atonia, there is still a debate regarding the source of this glycinergic neurotransmission. We recently demonstrated that glutamatergic neurons within the pontine sublaterodorsal tegmental nucleus (SLD) generate muscle atonia during PS and send descending inputs to the ventromedial medullary reticular formation (vmM) in rats[12]. Within the vmM, they contact glycine neurons that send monosynaptic inputs to spinal motoneurons[13]. Interestingly, the vmM also contains GABA cells expressing c-Fos after PS hypersomnia and spinally projecting neurons with a firing activity selective to PS[14,15]. Injection of glutamatergic agonists into the vmM induces muscle atonia, whereas neurotoxic lesion within this region produces an increased muscle tone associated with motor behaviors during PS[16,17]. According to these data, we thus proposed that GABA/glycine vmM neurons might be responsible for the muscle atonia during PS through the inhibition of somatic motoneurons[18,19]. This hypothesis has been challenged by Lu et al.[20] who found that large neurochemical lesions of the ventral medulla have no effect on atonia during PS. The same group later reported that smaller lesions in the same area induce an intermittent loss of atonia with exaggerated muscle twitches during PS[21]. Moreover, muscle tone during PS is reported to be unaffected after either optogenetic inhibition of GABA neurons within the ventral medulla or the removal of GABA/glycine neurotransmission from the vmM in GAD2-cre and vGAT$^{flox/flox}$ mice, respectively[21,22]. However, inactivating GABA/glycine signaling in cervical spinal cord provokes jerking movements in upper body territories during PS, suggesting a contribution of spinal interneurons in PS-related muscle atonia[23].

To make a significant step forward in this debate, we combined anatomical approaches to identify glycine neurons projecting to lumbar motoneurons that express c-Fos during PS hypersomnia in rats. Here, we show that such neurons were exclusively located in the vmM, not the spinal cord. We then studied the effects of genetic inactivation of GABA/glycine neurotransmission in vmM after the local knockdown of vGAT, the vesicular transporter of GABA/glycine necessary for their synaptic release and vesicle reloading[24]. Combining the use of short-hairpin RNAs against vGAT with innovative behavioral analyses, we demonstrate that impairment of GABA/glycine vmM neurotransmission in the rat is sufficient to mimic the major symptoms of human RBD. Notably, we validate a pre-clinical RBD model providing new opportunities for clinical research to improve patient treatment and to study mechanisms responsible for medication-induced RBD, as with antidepressants.

## Results

**Brainstem distribution of PS-activated glycine neurons**. The exact location of PS-on inhibitory pre-motoneurons within either the vmM or spinal cord remains to be clearly established. In an attempt to solve this issue, we performed three complementary anatomical-functional experiments in different groups of rats using c-Fos as a marker of neuronal activity. We first compared the distribution of glycine neurons, labeled by in situ hybridization (ISH) of glycine transporter 2 mRNA (GlyT2) that express c-Fos in the lower brainstem and lumbar cord (7–10 sp Rexed's layers) of PS recovery (PSR, $n = 6$), PS-deprived (PSD, $n = 5$), STEP ($n = 4$), and PS control (PSC, $n = 4$) rats (Fig. 1a; Supplementary Table 1 and Supplementary Fig. 1). In line with our previous studies using the same PS deprivation paradigm[14,25–28], PSR rats experienced significantly higher amounts of PS (38.8 ± 2.4%) during the last 150 min before sacrifice than PSD (0.4 ± 0.3%, Mann–Whitney $U$ test, $Z = −2.739$, $p = 0.006$) and PSC (12.2 ± 1.8%, Mann–Whitney $U$ test, $Z = −2.558$, $p = 0.01$) animals, due to both a significant increased number (23.8 ± 2.8 vs. 15.5 ± 5.0) and average duration of PS episodes (2.7 ± 0.4 vs. 1.5 ± 0.3 min) in PSR compared to PSC rats. Moreover, the lateral and dorsal paragigantocellular nuclei (LPGi and DPGi), ventral and alpha gigantocellular nuclei (GiV and GiA), and raphe magnus (RMg) contained significantly higher numbers of c-Fos+ neurons in PSR than PSD and PSC animals (Table 1). Further, we observed significantly higher numbers of c-Fos+/GlyT2+ neurons in PSR than PSC and PSD conditions in these nuclei (Mann–Whitney $U$ test, $p < 0,05$) where most of the c-Fos+ neurons were GlyT2+ (up to 82.5% for GiV; Fig. 1b, c; Table 2). In contrast, significantly fewer c-Fos+/GlyT2+ neurons were observed in the LPGi and GiV in STEP rats compared to PSR animals. Indeed, very few double-stained neurons were observed in these two structures in STEP rats (Mann–Whitney $U$ test, $Z = −2.558$, $p = 0.01$; Fig. 1a; Table 1).

At spinal levels, a significantly higher number of c-Fos+/GlyT2+ neurons was observed in PSD and PSR rats compared to PSC rats (Mann–Whitney $U$ test, $Z = −2.021$, $p = 0.04$). However, there was no significant difference between PSD and PSR groups (Mann–Whitney $U$ test, $Z = −0.577$, $p = 0.56$, Table 1). In contrast, a very high and significantly superior number of c-Fos+/GlyT2 neurons were observed in the spinal cord of STEP rats compared to the three other experimental groups (Mann–Whitney $U$ test, $Z = −2.309$, $p = 0.02$; Fig. 1a; Table 1). Double-labeled neurons constituted ≈77% of the c-Fos+ neurons in STEP rats, significantly higher than for the PSC (0%), PSD (31.6%) and, of particular importance, the PSR (14.8%) groups (Mann–Whitney $U$ test, $Z = −2.309$, $p = 0.02$; Fig. 1d).

These new functional data suggest that glycine inhibitory neurons that are specifically activated during PS hypersomnia are localized within the vmM, including the GiV, RMg, and GiA nuclei. Moreover, the activation of spinal glycine interneurons is likely independent of the vigilance states and related, instead, to locomotor activity.

The second step was to determine whether these inhibitory vmM neurons activated during PS project to the spinal somatic motoneurons. Thus, retrogradely labeled cells expressing c-Fos during PS hypersomnia were mapped in rats previously injected with fluorogold (FG) into lumbar motoneurons (Fig. 2a; Supplementary Table 1). In agreement with previous tract-tracing studies in rats[13], numerous FG+ neurons were found in the vmM nuclei with a marked ipsilateral dominance (Fig. 2b, c, e; Table 2). A large number of c-Fos/FG double-labeled neurons were counted, especially in the GiV and to a minor extent the RMg (Fig. 2b, c, f) and adjacent GiA (Fig. 2e). This highly contrasts with the lumbar spinal cord where only rare c-Fos+/FG+ neurons were observed despite the very high number of FG+ neurons found in layers 7 and 8 (Fig. 2d). These data indicate that descending, likely inhibitory, pathways emanating from vmM are recruited during PS.

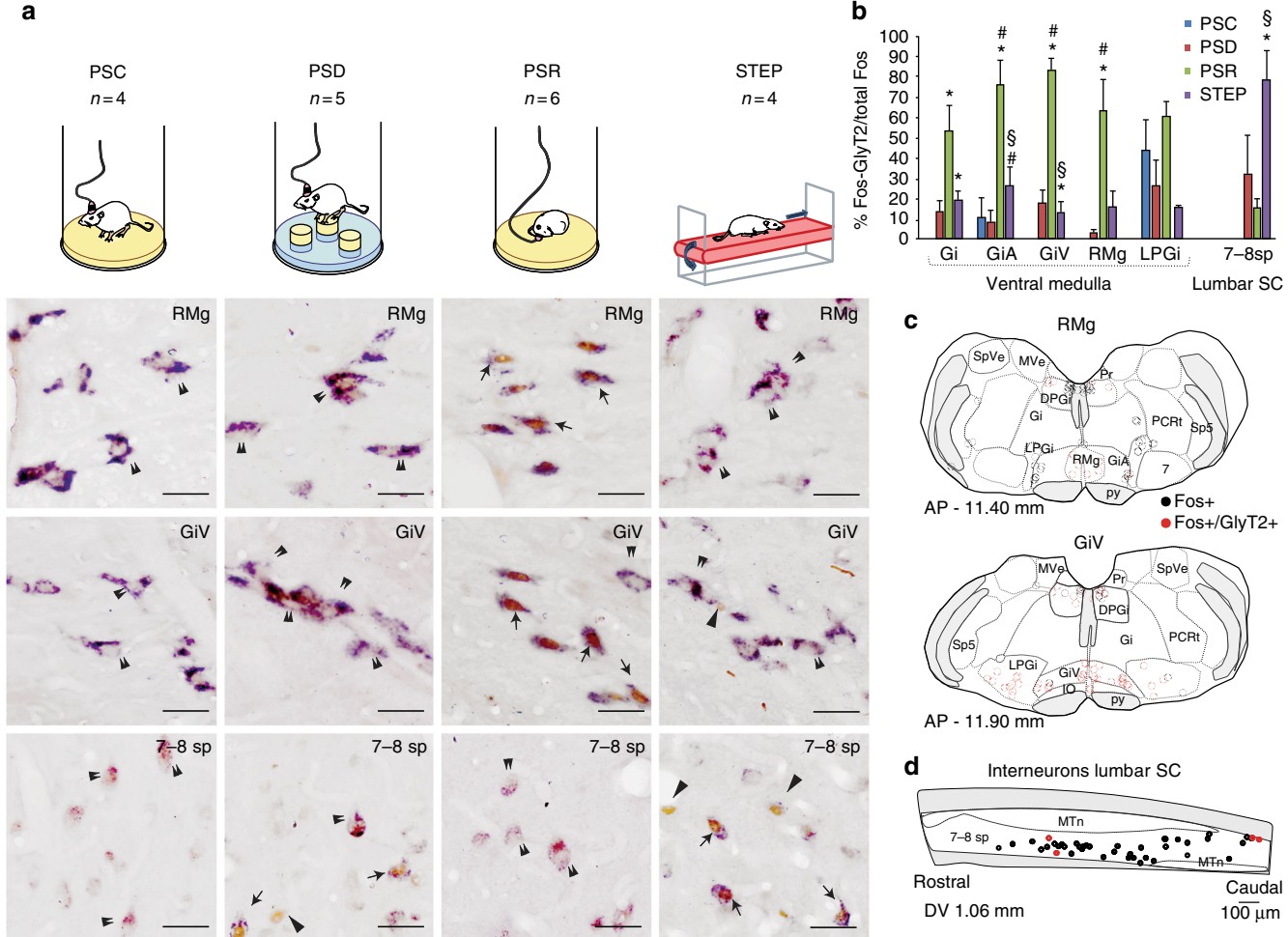

**Fig. 1** GABA/glycine neurons of the ventromedial medulla (vmM) are specifically active during PS rebound. **a** Photomicrographs comparing the distribution in RMg (upper row), GiV (middle row) and 7–8 sp Rexed's layer of lumbar spinal cord (SC, lower row) of GlyT2+ neurons expressing c-Fos in control (PSC, first column) and in rats submitted to a PS deprivation (PSD, second column), a PS rebound (PSR, third column) or a protocol of forced locomotion (STEP, fourth column). Neurons expressing c-Fos were colored in brown (nuclear staining, arrowheads) whereas GlyT2+ neurons were colored in blue (cytoplasmic staining, double arrowhead). Notice that the number of double-labeled neurons c-Fos+/GlyT2+ (arrows) in RMg (top line) and GiV (middle line) is higher during PSR compared to the 3 other experimental conditions. At spinal level, numerous c-Fos+/GlyT2+ neurons were found quite only in STEP rats. Scale bars: 50 μm. **b** Histograms (mean ± SEM; $n = 4$ PSC, $n = 5$ PSD, $n = 6$ PSR, $n = 4$ STEP) illustrating the percentage of c-Fos+ that were also GABA/glycine in nature (c-Fos+/GlyT2+) encountered within the vmM and lumbar SC for each experimental condition. **c, d** Drawings of sections showing the distribution of single c-Fos+ (black dots) and double-labeled c-Fos+/GlyT2+(red dots) neurons in the same representative PSR rat within the RMg and GiV (**c**, frontal sections) and lumbar SC at T13-L2 levels (**d**, horizontal section). Kruskal–Wallis tests followed by Mann–Whitney U tests *$p < 0,05$ compared to PSC, #$p < 0,05$ compared to PSD, §$p < 0,05$ compared to PSR. 7 facial nucleus, 7–8 sp 7–8 Rexed's spinal cord layers, Amb ambiguous nucleus, DPGi dorsal paragigantocellular nucleus, Gi gigantocellular reticular nucleus, GiA gigantocellular reticular nucleus (pars alpha), GiV gigantocellular reticular nucleus (pars ventral), IO inferior olivary complex, LPGi lateral paragigantocellular nucleus, MTn motoneurons, MVe medial vestibular nucleus, PCRt parvicellular reticular nucleus, Pr praepositushypoglossi nucleus, py pyramidal tract, RMg raphe magnus nucleus, SLD pontine sublaterodorsal tegmental nucleus, sp5 trigeminal spinal tract, SpVe spinal vestibular nucleus

The third step was to confirm that inhibitory vmM neurons send direct efferent projections to lumbar motoneurons. We took advantage of the expression of the reporter-protein mCherry in parent axons of transduced vmM neurons in the spinal cord after local injection of AAV-shRNA (see physiological study below). Indeed, dense plexuses of mCherry+ varicose anterogradely labeled fibers were observed closely apposed on soma of lumbar motoneurons immunostained for choline acetyltransferase (ChAT; Fig. 2g).

Taken together, our data indicate that the vmM contains numerous GABA/glycine neurons that send inhibitory mono-synaptic inputs to lumbar motoneurons and that are specifically recruited during PS. These neurons likely appear the best candidate for inducing the hyperpolarization of brainstem and spinal motoneurons underlying muscle atonia during PS. To functionally test this hypothesis, we inactivated inhibitory vmM neurons by local injections of AAV-shRNA targeting vGAT, the vesicular GABA/glycine transporter in freely moving adult rats.

**Efficient knockdown of vGAT mRNA and protein with shRNA.**
To check the specificity and efficiency of shRNA after 30 days of survival, we compared in Ctrl-shRNA and vGAT-shRNA rats the expression of vGAT mRNA within AAV injection sites and that of vGAT protein in anterogradely labeled fibers within lumbar cord (Fig. 3a). In control animals, a strong vGAT mRNA expression (Fig. 3h, i) was observed in neurons within the AAV injection sites delineated by the mCherry fluorescence (Fig. 3c–e).

**Table 1 Numbers of glycine neurons that expressed c-Fos in the medullary reticular formation and lumbar cord after PS recovery**

| Total Fos+ | | | | | |
|---|---|---|---|---|---|
| Medulla | n | PSC n=4 | PSD n=5 | PSR n=6 | STEP n=4 |
| DPGi | 10 | 2.5 ± 1 | 17.4 ± 8.5* | 101 ± 38.4*# | 41.3 ± 11.9* |
| Gi | 12 | 0.5 ± 0.3 | 30.2 ± 11.2 | 39.2 ± 15* | 124.5 ± 23.7* |
| GiA | 4 | 1.8 ± 1.2 | 9.8 ± 5 | 60.7 ± 35.3* | 34.8 ± 6.2* |
| GiV | 8 | 1 ± 0.7 | 25 ± 13.2* | 106 ± 28.1*# | 56.8 ± 12.4* |
| PCRT | 12 | 3.8 ± 1.3 | 106.8 ± 56.5* | 55 ± 17.2* | 47.3 ± 12.3* |
| RMg | 2 | 1 ± 0.6 | 14.4 ± 4.5* | 26 ± 11.2 | 26.3 ± 3.8*# |
| RPa | 6 | 0.5 ± 0.3 | 15.4 ± 9.3 | 10 ± 5* | 41.5 ± 14.5*§ |
| ROb | 6 | 1.8 ± 1.4 | 8 ± 4.8 | 6.7 ± 1.5 | 1 ± 0.7 |
| LPGi | 12 | 8.8 ± 3.5 | 89.4 ± 24.8* | 220 ± 39.8*# | 259.5 ± 25.1*# |
| Spinal cord interneurons | | | | | |
| 10 sp | 3 | 2 ± 0.9 | 33 ± 7.8 | 90 ± 50.8 | 61.8 ± 17.7 |
| 7–8 sp | 5 | 49.3 ± 6.4 | 313.3 ± 104.6* | 639.5 ± 255.6* | 1200 ± 295.4*# |
| **Fos+GlyT2+** | | | | | |
| Medulla | n | PSC n=4 | PSD n=5 | PSR n=6 | STEP n=4 |
| DPGi | 10 | 0.3 ± 0.3 | 3 ± 1.4* | 11.7 ± 2.2* | 5.5 ± 2.8 |
| Gi | 12 | 0 ± 0 | 4 ± 1.6 | 28.3 ± 10.8* | 23.8 ± 9* |
| GiA | 4 | 0.5 ± 0.5 | 0.6 ± 0.4 | 20.8 ± 8.7*# | 9.8 ± 4* |
| GiV | 8 | 0 ± 0 | 2.8 ± 0.9* | 80.5 ± 17.3*# | 7.5 ± 3.5*§ |
| PCRT | 12 | 1.8 ± 0.6 | 8.2 ± 5.2 | 11.2 ± 3 | 9.5 ± 5.5 |
| RMg | 2 | 0 ± 0 | 0.4 ± 0.2 | 16.7 ± 6.4*# | 3.8 ± 2.3 |
| RPa | 6 | 0 ± 0 | 0.2 ± 0.2 | 1.2 ± 0.5 | 0 ± 0 |
| ROb | 6 | 1 ± 1 | 4 ± 2 | 0.7 ± 0.3 | 0 ± 0 |
| LPGi | 12 | 4.3 ± 1.5 | 4.8 ± 1.7* | 74.5 ± 23*# | 37.5 ± 9.7*§ |
| Spinal cord interneurons | | | | | |
| 10 sp | 3 | 0 ± 0 | 4 ± 2 | 1.8 ± 0.8 | 9 ± 3.7 |
| 7–8 sp | 5 | 12 ± 4.7 | 85.3 ± 33.4* | 68 ± 5.4* | 533 ± 133.8*#§ |
| **%Fos+GlyT2+/total Fos+** | | | | | |
| Medulla | n | PSC n=4 | PSD n=5 | PSR n=6 | STEP n=4 |
| DPGi | 10 | 8.3 ± 8.3 | 23.3 ± 8.8 | 19.6 ± 4.6 | 10.3 ± 4.3 |
| Gi | 12 | 0 ± 0 | 13.1 ± 5.6 | 52.9 ± 12.4* | 18.7 ± 4.6* |
| GiA | 4 | 10 ± 10 | 7.9 ± 5.7 | 75.1 ± 12.2*# | 25.6 ± 9.1§ |
| GiV | 8 | 0 ± 0 | 17.1 ± 6.6* | 82.5 ± 5.9*# | 12.6 ± 5.4*§ |
| PCRT | 12 | 35 ± 11.9 | 7.4 ± 1.4 | 22.7 ± 2.6 | 17.8 ± 6 |
| RMg | 2 | 0 ± 0 | 2.8 ± 1.7 | 62.4 ± 15.3*# | 15.3 ± 7.9§ |
| RPa | 6 | 0 ± 0 | 0.4 ± 0.4 | 24.7 ± 15.9 | 0 ± 0 |
| ROb | 6 | 16.7 ± 16.7 | 37.4 ± 18.4 | 8.1 ± 4 | 0 ± 0 |
| LPGi | 12 | 40.7 ± 14.7 | 24.3 ± 12.0 | 56.7 ± 9.1 | 14.0 ± 3.3 |
| Spinal cord interneurons | | | | | |
| 10 sp | 3 | 25 ± 25 | 14.4 ± 5.8 | 13.4 ± 3.9 | 13.9 ± 2.3 |
| 7–8 sp | 5 | 0 ± 0 | 31.6 ± 18.8 | 14.8 ± 4.6 | 77.7 ± 14.1*§ |

Total number of c-Fos+ neurons (mean ± SEM), double-labeled c-Fos+GlyT2+ and percentage of double-labeled neurons in control (PSC, n = 4), PS-deprived (PSD, n = 5) and PS-rebound (PSR, n = 6) rats or animals forced to walk on treadmill (STEP, n = 4). For each rat and each nucleus considered, sums of labeled neurons were calculated on all consecutive sections (indicated by n, 150 µm interval) and then averaged for each rat's sample. The percentages displayed correspond to the ratio double vs. c-Fos-labeled neurons. Non-parametric Kruskal–Wallis test and unpaired Mann–Whitney U test
*p < 0.05 compared to PSC, #p < 0.05 compared to PSD, §p < 0,05 compared to PSR

In contrast, no cellular expression for vGAT mRNA was still visible within the injection sites of vGAT-shRNA rats compared to bordering regions not invaded by AAV spreading (Fig. 3b, f, g). Next, we observed in Ctrl-shRNA rats that a high number of anterogradely labeled mCherry+ fibers in close apposition to lumbar motoneurons co-expressed vGAT protein (Fig. 3i, left column). In vGAT-shRNA rats, mCherry+ fibers were similarly distributed, but there were less vGAT+ fibers and no apparent co-existence (Fig. 3j, right column). Thus, transduced inhibitory neurons in vGAT-shRNA rats no longer expressed the native vGAT protein and, therefore, were likely unable to release GABA/glycine at their synaptic terminals.

**Effects on sleep of vGAT inactivation in vmM neurons.** Only animals with AAV injections encompassing the entire vmM (namely the entire rostro-caudal extend of RMg and GiV bilaterally) were considered further for this analysis (vGAT-shRNA, n = 7; Ctrl-shRNA, n = 5; Fig. 3b, c). Animals with incomplete vmM transfection or with a spread of AAV encroaching contiguous vmM areas as LPGi and Gi were excluded since both areas do not contain inhibitory neurons recruited during PS projecting to lumbar motoneurons (see above). In both groups, vigilance states were quantified at day 30 after AAV injections (Fig. 4a). Daily amounts of waking (W) and slow-wave sleep (SWS) were not significantly different between Ctrl-shRNA and vGAT-shRNA rats while non-significant decreases in PS quantities (118.7 ± 4.6 vs. 98 ± 5.9 min; Supplementary Table 2) and percentage (8.2 ± 0.3% vs. 6.8 ± 0.4%; Mann–Whitney U test, Z = −1.949, p = 0.05) were noted. This was due to a significantly shorter mean duration of PS episodes compared to control animals (1.6 ± 0.1 vs. 1.3 ± 0.04; Mann–Whitney U test, Z = −2.680, p = 0.007; Supplementary Table 2; Fig. 4b) whereas the daily number of episodes was unchanged (76.2 ± 4.3 vs. 76.4 ± 4.3; Mann–Whitney U test, Z = −0.650, p = 0.5). The decrease in episode duration is likely due to frequent awakenings induced by abnormal movements displayed during PS in the experimental group (see below). Finally, no modification of the normalized EEG power spectrum was observed during PS, SWS, and W between the two groups (Fig. 4c).

**Table 2 Numbers of vmM neurons retrogradely labeled from lumbar cord expressing c-Fos during PS recovery**

| | n | FG+ | | Fos+FG+ | | %Fos+FG+/total FG+ | | %Fos+FG+/total Fos+ | |
|---|---|---|---|---|---|---|---|---|---|
| | | ipsi | contra | ipsi | contra | ipsi | contra | ipsi | contra |
| Gi | 6 | 69 ± 15 | 22 ± 5.8 | 4.5 ± 3 | 3 ± 2.3 | 8.9 ± 7.2 | 9.3 ± 5.8 | 8.9 ± 5.2 | 4.1 ± 1.4 |
| Giα | 2 | 24.5 ± 5.5 | 2.5 ± 1.3 | 1.8 ± 0.6 | 0.3 ± 0.3 | 8.8 ± 3.7 | 4.2 ± 4.2 | 7.5 ± 2.7 | 0.6 ± 0.6 |
| GiV | 4 | 144.8 ± 41.5 | 58 ± 16.2 | 43.3 ± 8.9 | 17.8 ± 4.4 | 31.9 ± 3 | 31.3 ± 2.5 | 42.6 ± 4.9 | 21.4 ± 2.8 |
| RMg | 2 | 35 ± 9.3 | 35 ± 9.3 | 7 ± 3.7 | 7 ± 3.7 | 24.3 ± 11 | 24.3 ± 11 | 19.7 ± 8.3 | 19.7 ± 8.3 |

Numbers (mean ± SEM) of single FG+, double-labeled c-Fos+/FG+neurons counted in the different medullary nuclei forming the vmM after PS rebound (PSR n = 4) rats. The percentages displayed correspond to the ratio double vs. single labeled neurons with each given marker of interest (FG+ or c-Fos+). For each rat and each nucleus considered, sums of labeled neurons were calculated on all consecutive sections (indicated by n, 150 µm interval) and then averaged
Gi, gigantocellular reticular nucleus; GiA, gigantocellular reticular nucleus, pars alpha; GiV, gigantocellular reticular nucleus, pars ventral; RMg, raphe magnus nucleus

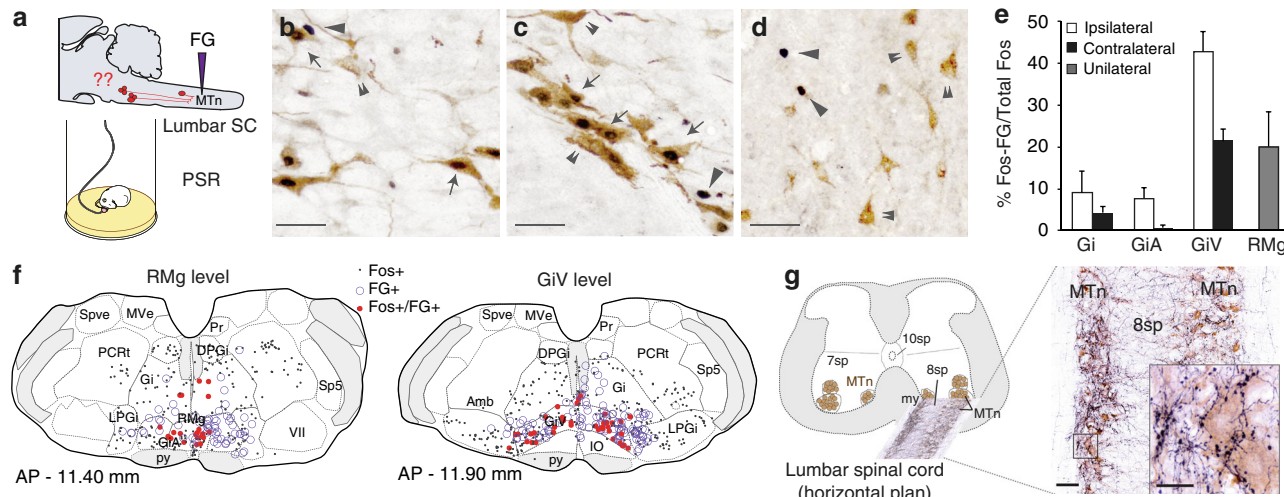

**Fig. 2** Neurons of the vmM specifically activated during PS send direct projections to the lumbar motoneurons. **a** Drawing illustrating the experimental paradigm with a FG injection in lumbar motoneurons at spinal L1–L2 levels in rats allowed to sleep recover (PSR) after a 72-h deprivation of PS using the flower-pot method (n = 4). **b–d** Photomicrographs at high magnification of sections at the level of RMg (**b**), GiV (**c**), and 7–8 sp Rexed's layers at lumbar SC (**d**), double immunostained for c-Fos and FG in a representative PSR rat. Neurons expressing c-Fos were colored in black (nuclear staining, arrowheads) whereas FG+ neurons were colored in brown(cytoplasmic staining, double arrowhead). Notice the massive numbers of double-labeled neurons c-Fos+/FG + (arrows) in RMg and GiV compared to lumbar SC where scattered, if any c-Fos+/FG+ cells are seen after PS recovery. **e** Percentage of c-Fos+ neurons in the vmM that are retrogradely labeled after a FG injection in the lumbar SC (PSR n = 4, values are mean ± SEM). **f** Drawings of two frontal medulla sections illustrating the bilateral distribution of single FG+ (violet circles), single c-Fos+ (black dots), and double-labeled c-Fos+/FG+ neurons (red dots) in the RMg and GiV in the same PSR rat. **g** Photomicrograph overview of one horizontal lumbar section at the level of 7–8 sp Rexed's layers showing fibers emanating from transducted GABA/glycine vmM neurons with AAV as immunolabeled for mCherry (black fibers). Notice the very high density of mCherry immunolabeled fibers surrounding the soma of lumbar motoneurons (brown cytoplasmic staining), often in close apposition with them suggesting that vmM neurons have direct synaptic contacts with somatic motoneurons. Scale bars: 40 µm in **b–d**; 50 µm in **g** overview and 20 µm in enlarged square

**Inhibitory vmM neurons are essential for PS atonia**. In Ctrl-shRNA rats, PS is characterized by a reduced muscle tone relative to the preceding SWS episode (Fig. 4 d,f) as confirmed by the mean EMG PS/SWS ratio of 0.97 ± 0.03 calculated on nuchal EMG (Fig. 4 g). In contrast, the muscle tone in vGAT-shRNA rats was increased during all PS episodes (92.5 ± 7.5% with an increased total motor activity; Fig. 4e,f) and reflected by a mean EMG PS/SWS ratio of 1.36 ± 0.04, significantly superior to 1 and to that of Ctrl-shRNA rats (Mann–Whitney U test, Z = −2.680, p = 0.007; Fig. 4g). Importantly, no difference was seen between groups in the waking and SWS EMG activities (Mann–Whitney U test, Z = −0.893, p = 0.37 and Z = −1.543, p = 0.12, respectively; Fig. 4f), suggesting that physiological effects are specific to PS. These data indicate that a state of PS without atonia was induced in rats after the genetic inactivation of GABA/glycine neurotransmission from vmM neurons.

By visual scrutiny of sleeping rats in their home barrels, we noticed that vGAT-shRNA animals displayed an increased motor activity during PS (vs. control rats), especially in distal extremities (tail, limbs, whiskers, ears) that are not reflected by nuchal EMG. Thus, we objectively quantified body movements during PS in both groups by an offline actimetric analysis of videos time-locked to polysomnographic recordings. We found that motor events are frequent during PS since present in 79.7 ± 5.3% of episodes (Fig. 5h). Further, the amounts of movements during PS (Mann–Whitney U test, Z = −2.842, p = 0.004; Fig. 5a–c) and the mean number of motor events per min (7.3 ± 5.5 vs. 0.8 ± 0.1; Mann–Whitney U test, Z = −2.842, p = 0.005; Fig. 5d) were increased in vGAT-shRNA compared to Ctrl-shRNA animals. The mean duration of these motor events was also significantly increased compared to naturally occurring twitches in Ctrl-shRNA rats (0.49 ± 0.08 vs. 0.3 ± 0.03 sec; Mann–Whitney U test, Z = −2.842, p = 0.005; Fig. 5e). As a consequence, the percentage of PS with movements was significantly higher in vGAT-shRNA compared to Ctrl-shRNA animals (6.1 ± 0.2 vs. 0.6 ± 0.1; Mann–Whitney U test, Z = −2.842, p = 0.005; Fig. 5f). No

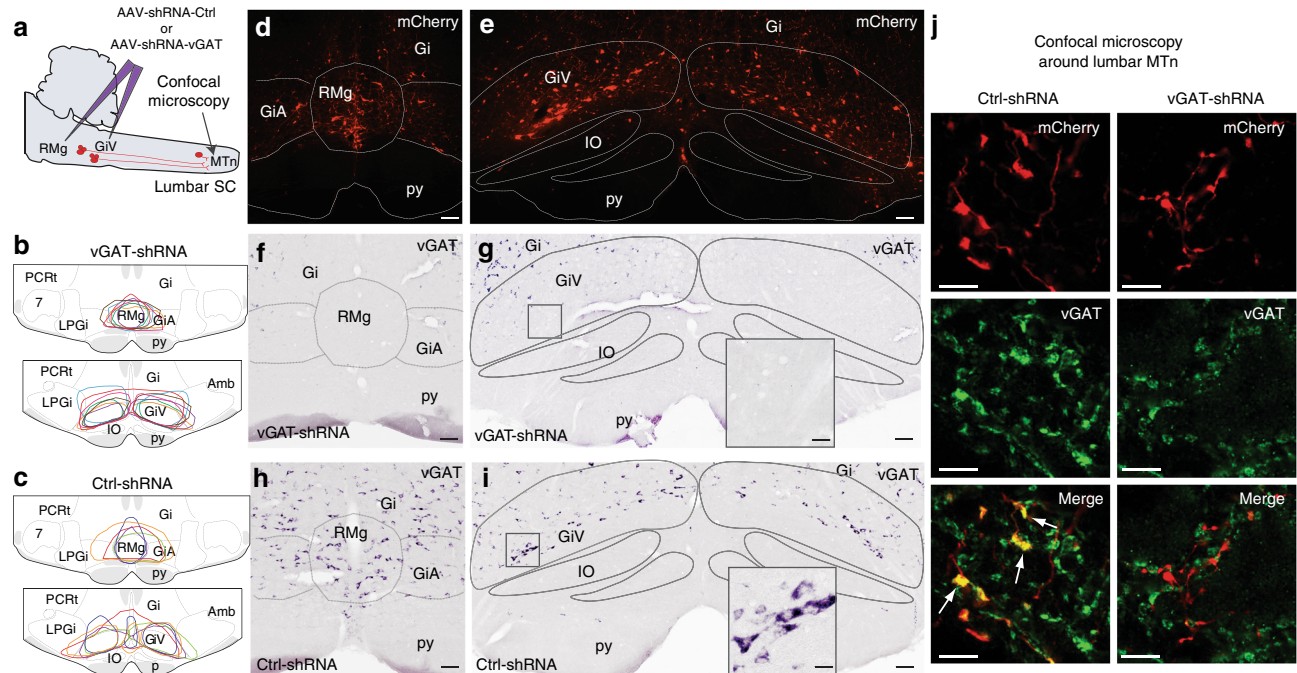

**Fig. 3** Verification of the genetic inactivation of endogenous vGAT mRNA and native vGAT protein in GABA/glycine vmM neurons. **a** Scheme of the in vivo experiments: AAV-shRNA injections were made in three points to genetically inactivate GABA/glycine neurons present bilaterally in the GiV, and GiA and the midline RMg and to trace anterogradely their efferent projections of transduced neurons until the lumbar spinal cord. **b**, **c** Drawings reporting the location and the extent of AAV injection sites at day 30 post-injection for each Ctrl-shRNA (**b**, n = 5) and vGAT-shRNA (**c**, n = 7) treated rat considered for the physiological study. The injection spreading was demarcated by the spontaneous mCherry fluorescence. Notice that for these rats, the AAV injections covered the largest part of the vmM and avoided the LPGi laterally and Gi dorsally. **d**, **e** Low power photomicrographs of the RMg/GiA (**d**) and the GiV (**e**) in a representative control rat showing the high number of transduced neurons after AAV injection. **h**, **i** Photomicrographs of adjacent sections treated for vGAT mRNA ISH. Notice the high number of strongly labeled neurons in RMg, GiA, and GiV indicating that the expression of vGAT mRNA is normal. **f**, **g** Photomicrographs of vmM sections taken from a representative vGAT-shRNA rat after the ISH labeling of vGAT mRNAs. Note the absence of labeled neurons in RMg, GiA, and GiV within the injection site. **j** Confocal photomicrographs comparing in representative Ctrl-shRNA (left) and vGAT-shRNA (right) rats the expression in lumbar motoneurons pool of the native vGAT protein (green, middle panels) in synaptic terminals emanating from transduced vmM neurons (mCherry, top panels). As evidenced on merged photomicrographs, the vGAT protein is expressed in axons and synaptic terminals (arrows) emanating from vmM neurons in Ctrl-shRNA rats (yellow, bottom left panel) but is virtually absent in vGAT-shRNA rats (bottom right panel). Scale bars: 100 μm in **b**, **c**; 40 μm in enlarged squares; 5 μm in **f**

modification of the actimetry during SWS was observed in both groups of animals (Mann–Whitney $U$ test, $Z = -0.244$, $p = 0.8$; Wilcoxon signed rank test, $Z = -1,014$, $p = 0,3$; Fig. 4c).

During the initial scoring of polysomnographic recording, we found that the increase in muscle tone and abnormal movements in treated rats are not sustained along PS episodes and across PS episodes. This was confirmed by the computation of EMG data from Ctrl-shRNA and vGAT-shRNA rats (Fig. 5g, h). Moreover, data extracted from each individual activity map showed that the actimetry level in vGAT-shRNA rats was not uniformly distributed during PS. It followed a temporal trajectory similar to that reported for natural muscle twitches during PS in the rat[29], still peaking during the last third of PS bouts (≈47% of abnormal movements compared to the first and second third of PS episode with ≈21 and ≈32%, respectively).This gradual increase along PS episodes is concomitant to an increase in time spent moving (from ≈4% of time for the first third to ≈11% for the last third; Fig. 5h).

In conclusion, the loss of muscle atonia during PS after the genetic inactivation of GABA/glycine vmM neurons facilitates the occurrence of abnormal and intense motor enactments in experimental rats.

**Dreaming motor behaviors during PS without atonia**. During PS, Ctrl-shRNA rats slept most of the time in the standard curled

position, displaying discrete movements (so-called "twitches"; Supplementary Movie 1). This position is often lost in vGAT-shRNA rats due to intense oneiric-like movements. Qualitatively, they correspond to assorted non-elaborated behaviors that are loosely arranged at the level of the head, fore and hindlimbs, tail, or nose, rarely synchronized over multiple body territories (Supplementary Movie 2). Intermittent complex, vigorous, and likely uncontrolled movements were also observed looking like seeking for food with the snout in woodchips, trying to run or jump. These violent movements often induced an awakening of vGAT-shRNA rats when they hit the barrel wall, likely explaining the shortened PS bouts (Supplementary Movies 3 and 4). During oneiric-like motor behaviors, rats maintained their eyes closed, indicating they are asleep. Finally, vGAT-shRNA rats depicted standard weight increases and normal locomotor activity during waking. Abnormal motor events were absent during SWS.

Taken together, our data indicate that inhibitory neurons of vmM are essential for muscle atonia during PS since their inactivation favors oneiric-like motor behaviors during a state of PS without atonia.

## Discussion

Despite decades of basic research, there is still an ongoing debate regarding the location of the GABA/glycine inhibitory pre-

 

motoneurons that produce muscle atonia during PS and that are turned on by excitatory inputs from pontine glutamate PS-on neurons within the SLD[12,30]. To date, two anatomical configurations of the network promoting PS atonia have been proposed: one locates the inhibitory relay between SLD and somatic motoneurons in the vmM[18,19], the other in 7–8 sp Rexed's layers in proximity to motoneuron pools[20,21,23]. Our goal was to confront both hypotheses using complementary anatomical, molecular, genetic, and functional approaches in rats. First, both anterograde and retrograde tract-tracing data convincingly show that GABA/glycine neurons in vmM send projections to lumbar motoneurons and are activated during PS. In contrast to spinal interneurons that appear primarily engaged during walking. Second, the genetic inactivation of GABA/glycine neurotransmission in vmM neurons is sufficient to disrupt muscle atonia during PS and to elicit abnormal oneiric motor behaviors without any effect during waking and SWS. These new data demonstrate that the GABA/glycine pre-motoneurons essential for muscle atonia during PS are located in the vmM rather than the spinal cord. Further, they validate a reproducible pre-clinical RBD model in rodents, providing a new translational research approach for disease management.

By combining the detection of c-Fos protein and GlyT2 mRNAs, we show in rats that the vmM is the unique brain area containing a high density of GABA/glycine neurons expressing c-Fos during PS rebound. More than 80% of these c-Fos+ neurons in RMg and GiV are inhibitory in nature. However, these neurons do not express c-Fos in rats deprived of PS or following forced locomotion. Our results are in agreement with c-Fos/glycine double immunostaining made in cats after induction of PS hypersomnia by carbachol injection in the pons[31]. We further report for the first time that a large proportion (40%) of the c-Fos+ cells found in the vmM after PS rebound send projections to lumbar motoneurons. These data are supported by previous studies showing that lumbar and cervical motoneurons receive direct synaptic inputs from GABA/glycine vmM neurons[13,32]. In contrast, only a small percentage of lumbar GlyT2+ neurons retrogradely labeled from lumbar motoneurons express c-Fos after PS rebound, whereas they are strongly activated during locomotion. This indicates that spinal pre-motoneurons are not significantly recruited during PS, in line with previous intracellular recordings showing that glycine Renshaw cells, known for tuning spinal motoneuron firing, are inactive during pharmacologically induced PS in cats[33]. Taken together, the present data strengthen our early hypothesis that GABA/glycine neurons in the vmM are the best pre-motoneuron candidates to convey hyperpolarizing inputs to the somatic motoneurons during PS[18,19].

We next injected AAV-shRNA targeting the GABA/glycine vesicular transporter to block constitutively inhibitory neurotransmission in vmM, and directly address its role in PS. Previous data have convincingly demonstrated the ability of this molecular method to knockdown the neuronal expression of targeted proteins in a long-term manner in the adult rat[12,24]. This targeted approach is superior to standard techniques such as electric stimulation, cytotoxic lesioning, or local pharmacology, techniques lacking the desired cellular selectivity[20,34,35]. Here we show that the sleep-waking cycle is not modified in experimental (vs. control) rats, although a non-significant reduction in PS quantities was observed (−18%) reflecting a shortened bout duration, likely due to the occurrence of violent movements often leading to premature awakening. Our physiological data indicate that GABA/glycine vmM neurons are not necessary for the generation of PS, but, instead, are critical for the control of PS muscle atonia. Contrary to our findings, no effect on muscle atonia was reported after genetic inactivation of GABA/glycine neurons in the medial

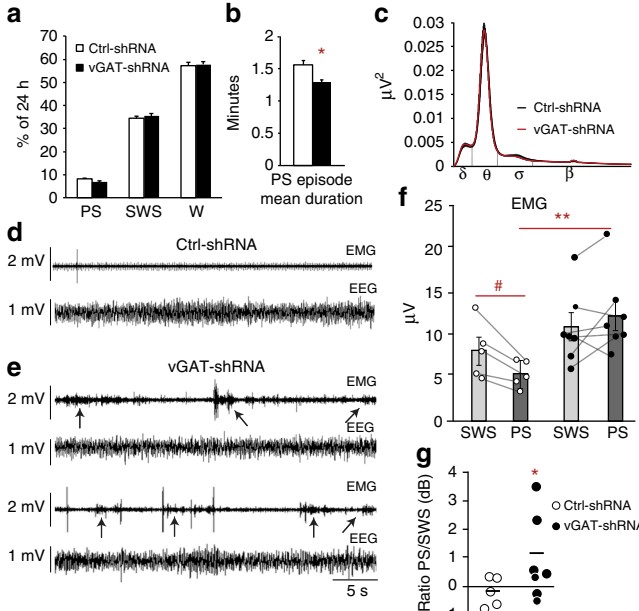

**Fig. 4** Loss of muscle atonia during PS after the genetic inactivation of GABA/glycine vmM neurons. **a, b** Histogram comparing the daily percentages of W, SWS, and PS in Ctrl-shRNA (open bars) vs. vGAT-shRNA (filled bars) rats at day 30 after AAV injections. Notice the significant reduction of the PS episode duration in vGAT-shRNA vs. Ctrl-shRNA rats (**b**). **c** Normalized mean power spectrum in a control (black) and experimental (red) rats showing no differences in EEG during PS between both groups. **d, e** Raw polysomnographic recordings during PS of two representative Ctrl-shRNA (**d**) and vGAT-shRNA rats (**e**) showing the loss, although irregular, of muscle atonia (arrows) concomitant to a strong increase of phasic twitches on nuchal EMG after the genetic inactivation of GABA/glycine vmM neurons. **f** Dot plots comparing mean EMG values during PS and SWS episodes (bars) in Ctrl-shRNA (open circles) vs. vGAT-shRNA (filled circles) rats. Dashed lines connect SWS with PS mean values for each animal. Control rats show an expected diminution of mean EMG values during PS compared to preceding SWS (i.e., PS atonia). In experimental rats, mean EMG values are comparable during PS and SWS unraveling the loss of atonia during PS. Interestingly, PS EMG is significantly increased in vGAT-shRNA rats compared to PS EMG in Ctrl-shRNA congeners. **g** Dot plots represented in decibels showing that PS/SWS ratio of mean EMG values is increased in experimental (filled circles) vs. control (open circles) rats. Mann–Whitney U tests, *p < 0,05 compared to Ctrl-shRNA; Wilcoxon signed rank test #p < 0,05 compared to mean EMG values during SWS

medulla (partly including the vmM) in vGAT[flox/flox] mice[21]. This discrepancy might be due to the location of AAV injections which were centered on the rostromedial medulla in the previous study (as illustrated in their Fig. 4c, d), mostly avoiding the GiV where the largest contingent of inhibitory neurons expressing c-Fos during PS is located.

Optogenetic and chemogenetic cell silencing in the ventral medulla of GAD2-cre mice has been shown to suppress PS without an effect on muscle tone or abnormal motor activity[22]. A difference in genetic targeting might explain the conflicting physiological data with our study since this prior work inactivated only GAD2-expressing neurons and not GABA and glycine neurons. Moreover, their AAV injections targeted the LPGi, encroaching on the lateral GiV but avoiding the medial GiV, RMg, and GiA (as illustrated in their Fig. 1a). In addition, these authors reported that transfected neurons project strongly to the locus coeruleus (LC) and ventrolateral periaqueductal gray

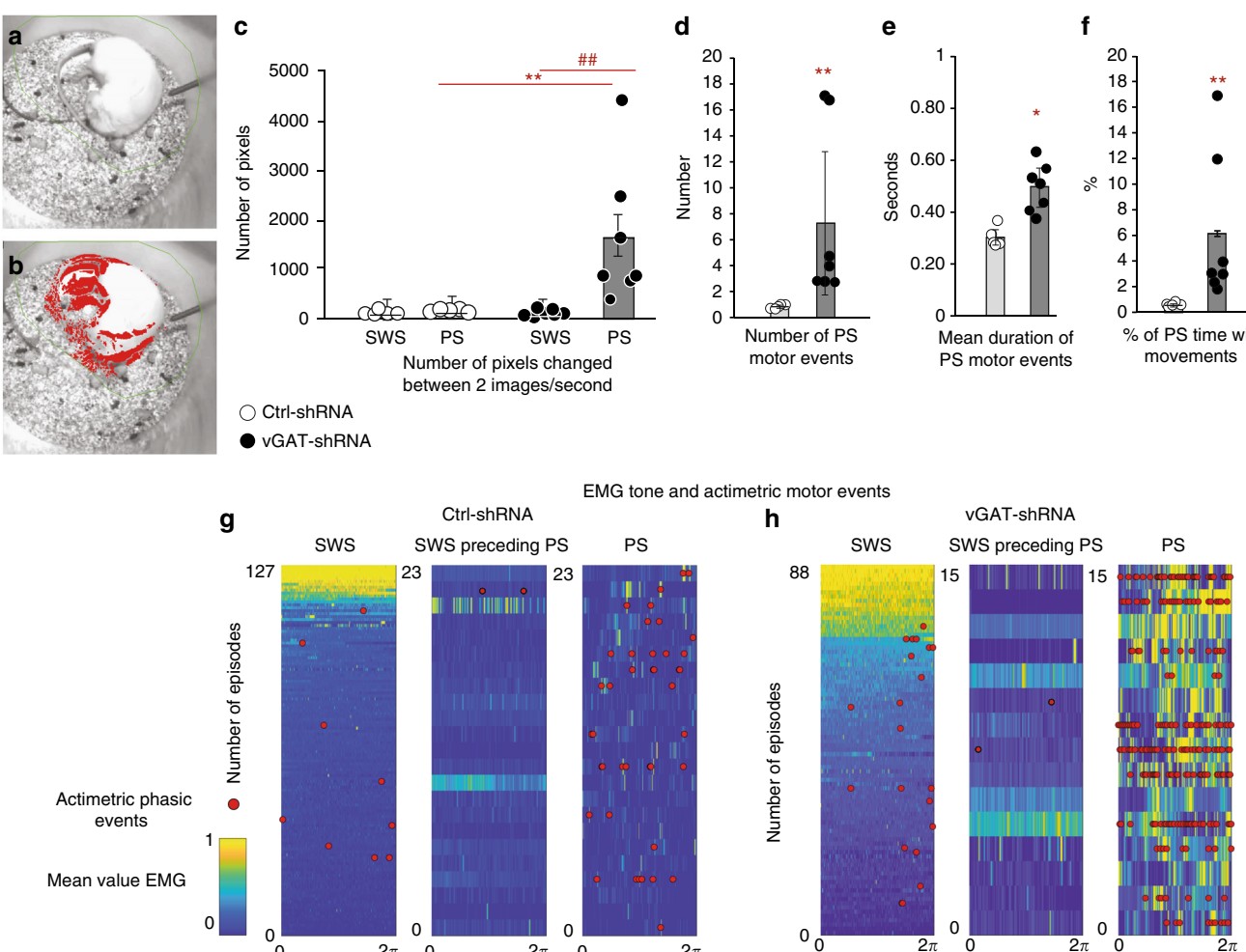

**Fig. 5** RBD-like behaviors are displayed during PS after the genetic inactivation of GABA/glycine vmM neurons. **a**, **b** Examples of captured video images of representative Ctrl-shRNA (**a**) and vGAT-shRNA rats (**b**), both during a PS episode. Each red point corresponds to a gray color changed pixel between two successive images (see Materials and Methods sections). The density and location of red points respectively reflect intensity of rat's movements and body territories where they occurred. **c** Histogram comparing in both rat groups the mean actimetry per second of SWS and PS calculated by counting the number of changed pixels between two successive video images due to animal movements. Oneiric movements appear only during PS in vGAT-shRNA rats. **d**, **e** Histograms comparing in bot rat groups the mean number of motor events per PS bout (**d**) and the mean duration of motor events (in seconds, **e**) per PS episode. **f** Histogram showing the percentage of PS time with movements in both rat groups. Note that all actimetry parameters calculated thanks to video recordings and that directly reflect the movements displayed by rats during PS are significantly higher in vGAT-shRNA vs. Ctrl-shRNA animals. Mann–Whitney $U$ tests, *$p < 0.05$; **$p < 0.005$ compared to Ctrl-shRNA. **g**, **h** Activity maps of one representative Ctrl-shRNA (**g**) and vGAT-shRNA rats (**h**) illustrating EMG nuchal changes and actimetric phasic events during SWS, SWS preceding PS, and PS. SWS and PS episodes are sorted vertically from the highest (top, yellow values) to the lowest (bottom, blue values) mean muscle tone combined with actimetry events. The SWS episodes preceding PS are sorted in the same order as their related PS episodes. The x-axis represents the episode duration normalized between 0 and $2\pi$. Notice the increase of muscle tone and actimetry motor events during PS in vGAT-shRNA animals and the higher probability of movement occurrence during the last third of PS bouts

(vlPAG). We previously demonstrated that the LPGi, not the vmM, contains GABA/glycine neurons activated during PS and projecting to LC and vlPAG[27,36,37]. Hence, it is clear that the LPGi contains ascending inhibitory PS-on neurons in position to inhibit the wake-active noradrenergic LC neurons during PS, potentially explaining the wake effects after their optogenetic/chemogenetic silencing[22]. Furthermore, our data clearly demonstrate that GABA/glycine PS-active neurons of the vmM (1) project to spinal motoneurons, (2) are critical for muscle atonia during PS since their genetic inactivation is sufficient to disrupt this state-specific control, and (3) are under direct excitatory inputs originating from glutamatergic SLD neurons, the pontine generator of muscle atonia during PS[12]. These findings de facto eliminate a primary role for spinal inhibitory pre-motoneurons in

PS atonia, supported also by their expression of c-Fos during forced locomotion but not during PS hypersomnia.

In addition to the loss of muscle atonia, the deletion of GABA/glycine transmission from vmM neurons favors aberrant motor behaviors during PS in experimental rats. Such abnormal movements never occurred during SWS. Moreover, EMG values, locomotion and different behavioral activities (e.g., grooming and exploring) remained normal during waking. In human polysomnography, the recording of several muscles is routinely performed for diagnostic purposes. Implementing multiple EMG recordings is difficult in rodents and nuchal EMG does not reflect movements in each body territory. To address this potential limitation, we validated two complementary offline analysis methods based on nuchal EMG signals and video recording to

objectively quantify changes in muscle tone and movements during PS in response to the genetic inactivation of glutamatergic SLD neurons[12] and GABA/glycinergic vmM neurons (present study). In both studies, experimental rats remarkably recapitulated the physio-pathological profile of RBD patients suffering from the loss of muscle paralysis during PS, as seen by the forceful motor enactments without a significant impact on PS quantities[1].

Our findings suggest similar anatomical-pathological substrates responsible for the abnormal behavioral expression during PS in both patients and our rat pre-clinical models. Indeed, RBD has been recorded in patients with inflammatory lesions of the dorsal pons or vmM region[38]. Functional neuroimaging and post-mortem brain studies show evidence for the presence of Lewy body pathology and neuronal loss in both areas in patients with Parkinson's disease and RBD[2,39]. There is also a reported case of α-synuclein pathology in the vmM of a 72-year-old male with idiopathic RBD who had a 15-year history of dream-enactment behavior[40]. In this latter patient, neither Parkinsonism nor dementia was ever detected during serial neurological examinations. Post-mortem neuropathological examination revealed Lewy body disease. As shown in Table 1 of that report, which lists the distribution of α-synuclein pathology and neuronal loss, the medullary reticular formation had moderate Lewy bodies and Lewy neurites but no neuronal loss. Although the anatomical location of identified pathology in this human study is consistent with our findings, the role of this pathology in disease manifestation remains unclear. Quantifying neuronal loss is more important than identifying α-synuclein pathology, particularly since the presence of a protein deposit does not necessarily mean indicate the cause of clinical symptoms[41].

It is also important to note that different symptomatic profiles have been described in human RBD, characterized by either PS without atonia (also known as REM without atonia, or RWA) alone or in combination with abnormal motor behaviors[42]. In our pre-clinical models (as shown here and in Valencia Garcia et al.[12]), rats with a subtotal genetic inactivation of either glutamatergic SLD or GABA/glycinergic vmM neurons present both types of profiles, although the vast majority of PS bouts depict a combination of RWA and high actimetry levels. It is thus possible that the dissociation reported in human RBD may reflect the severity and intensity of the inflammatory response, amount of α-synuclein deposited or number of destroyed SLD/vmM neurons. We hypothesize that the severity of symptoms may be linked with the degree of damage in one or both areas. It would be interesting to study the impact on RWA and RBD from different combinations of size vs. location of genetic inactivation using shRNA. It would be also of clinical relevance to test whether neurochemical mechanisms targeting SLD/vmM may explain medication-induced RBD observed in patients treated with selective serotonin reuptake inhibitors or Tricyclic antidepressants[42].

In a broader perspective, it is understood that ≈80% of patients suffering idiopathic RBD develop a synucleinopathy such as Parkinson's disease in a time window of 10–15 years[3–6,43]. Hence, RBD is considered a valuable prodromal marker of such neurodegenerative pathologies[44]. Behavioral similarities in the present treated rats and RBD patients strongly point out that damage to either glutamatergic SLD or GABA/glycinergic vmM neurons may underlie the human disorder. Aggregates of misfolded α-synuclein might initially target the vmM and then propagate in the caudo-rostral brain axis through axonal projections to the SLD[45]. Determining whether vmM neurons are indeed specifically targeted in human RBD is a challenge of major clinical relevance to understand the etiology of this disorder. Our reproducible pre-clinical rat model provides an experimental method to tackle these basic questions and to design alternative pharmacological treatments.

## Methods

**Animals.** Sprague Dawley male rats (240–280 g, Charles River Laboratories) were housed individually in Plexiglas barrels (30 cm diameter, 40 cm height) under a constant 12-h light/dark cycle and temperature (23 ± 1 °C). Pellets (A04 SAFE, Extra Labo) and water were available ad libitum.

**Surgical procedures.** Microiontophoretic ejection of Fluorogold (FG) in lumbar motoneurons: Under anesthesia (Ketamine/Xylazine, 100 and 50 mg/kg, Virbac), T13-L3 vertebras were exposed. A hole was drilled in the right L2 apophysis to lower (−2.2 mm) a glass micropipette (7–10 μm external tip diameter) backfilled with 1% FG solution (Sigma-Aldrich). Once connected to a CS4 current generator (Transkinetics), a constant current (+1 μA) was delivered for 15 min. These rats (n = 4) were sacrificed after 10 days of recovery.

Stereotaxic adeno-associated virus (AAV) infusion: To involve the whole vmM, anesthetized rats (n = 12) received injections (300 nl) bilaterally in the gigantocellular nucleus (GiV; AP, −13.5 mm Bregma; ML, ±1; DV, 8.5) and in the raphe magnus on medulla midline (RMg; AP, −13.2 mm; ML, 0; DV, 8.5 according to Paxinos and Watson atlas). This was done with an injection cannula (7° posterior angle, 33 gauge; Plastics One) filled with the solution of AAV-shvGAT-mCherry or AAV-shCtrl-mCherry and connected to a 10 μl syringe (Hamilton) placed into an UltraMicroPump with SYS-Micro4 controller (40 nl/min; WPI). After completion of the injection procedure, all rats were standardly prepared for polysomnography[14,46]. Briefly, four stainless-steel epidural electrodes (Anthogyr) were screwed to the skull over frontal (AP, +3 mm to Bregma; ML, 1), parietal (AP, −4; ML, 3), occipital (AP, −8; ML, 3), and cerebellar cortices (AP, −12; ML, 3; reference electrode). Two gold-coated electrodes were inserted in between neck muscles for a differential EMG recording. Electrode leads were finally connected to a miniature plug (Plastics One) and securely fixed to the skull using acrylic Superbond (Sun Medical Co) and dental Paladur cement (Heraeus Kuzler). Rats were sacrificed at day 30 post-surgery.

**Generation of viral vectors.** Methods for generating the AAV-shRNA have been detailed previously[12,24]. The shvGAT sequence used was TCGACGTCAA-GAAGTTTCCTA. The virus titer was $4.5 \times 10^{12}$ particles/ml for both AAV-shCtrl-mCherry and AAV-shvGAT-mCherry.

**Polysomnographic and video recording.** After recovery from surgery during 5–7 days, rats were connected to the acquisition set-up and continuously recorded. Unipolar EEG and bipolar EMG signals were amplified (MCP+, Alpha-Omega) and analog-to-digital (sampling rate 520.8 Hz) converted using a 1401 Plus interface (CED). For continuous video acquisition, we used digital black/white cameras (GigE PoE, 1200 × 900; Elvitec) managed by Streampix 6 (NorPix).

**Paradoxical sleep rebound and forced locomotion.** Before their sacrifice, rats dedicated to tract-tracing studies and mapping of active neurons were submitted to a protocol of PS deprivation (72-h) and recovery (2.5-h) with the standard flow-erpot method[12,28,47]. Three experimental sleep groups of rats were made: control (PSC, n = 4), PS deprivation (PSD, n = 5), and PS rebound (PSR, n = 4 for tract-tracing and n = 6 for c-Fos/GlyT2 experiments). During deprivation, food and water were available ad libitum and the barrels were cleaned daily. To compare the distribution of glycine neurons expressing c-Fos during PS and during locomotion, a fourth experimental group was shaped with rats (STEP, n = 4) trained to walk during 4 days with a daily incremental duration on a treadmill (Simplex II, Columbus Instruments; 6 m/min, 0° incline). The last training day, the rats were sacrificed after 120 min of forced walking.

**Histological procedures.** Preparation of sections: Under deep anesthesia with pentobarbital (150 mg/kg, ip, Ceva Santé Animal), rats were transcardially perfused with a solution of 4% paraformaldehyde[12,14,46]. Serial free-floating coronal sections (25 μm-thick for the tract-tracing studies and 30 μm-thick for ISH) were made from brainstem and horizontal sections (30-μm-thick) from lumbar cords (T13-L2).

c-Fos/FG and mCherry/ChAT double-immunostaining: Brainstem and lumbar sections were respectively incubated in rabbit antiserum against c-Fos (1:10000; cat #PC38, Merck Millipore) or rat antiserum against mCherry (1:100,000; cat #M11217, ThermoFisher) in PBST-Az (PBS containing 0.3% of Triton X-100 and 0.1% of Natriumazide) for 3 days at 4 °C. Sections were then treated according to the standard sequential protocol using biotinylated secondary antibodies (horse anti-rabbit or rabbit anti-rat IgG, 1:1000; cat #BA1100 and BA4000 respectively, Vector Labs), ABC elite kit (1:1000; cat #PK-6100, Vector Labs) and finally revealed with the DAB-Ni histological technique[12]. Then, c-Fos immunostained sections were incubated in a rabbit antiserum against FG (1:20,000; cat #AB153-I, Sigma-Aldrich) and mCherry immunostained sections in a goat antiserum against ChAT (1:5000; cat #AB144P, Chemicon). In this case, histological revelations were made in DAB solution without nickel[12].

ISH of vGAT mRNA: Antisense and sense digoxigenin-labeled probes against vGAT were synthesized using a nonradioactive RNA labeling kit according to manufacturer's instructions (Roche Diagnostic). The probe template consisted of a partial vGAT cDNA sequence, flanked by SP6 and T7 polymerase binding sequences, obtained by RT-PCR from a brainstem mRNA pool[46]. The ISH protocols for vGAT mRNA were similar to those previously validated[12,14,46].

Combined c-Fos immunostaining and GlyT2 mRNA ISH: The antisense and sense digoxigenin-labeled probes against mRNA for GlyT2 were synthesized as described above. Along these experiments, 0.2% RNase inhibitor (ProtectRNA, Sigma-Aldrich) was added to buffers. Sections from PSC, PSD, PSR, and STEP rats were first revealed for c-Fos immunohistochemistry using DAB-Ni technique. After rinses, they were then treated with GlyT2 probe as above for vGAT ISH.

Double mCherry/vGAT immunofluorescence: Free-floating lumbar sections were incubated simultaneously with rabbit IgG against vGAT (1:5000; cat #131003, Synaptic Systems) and rat IgG against mCherry (1:50,000) and then in a mixture of secondary antibodies tagged with Alexa Fluor 488 and 594 (1:500; cat #A21206 and A21209, respectively, ThermoFisher). Once mounted on slides coverslipped Fluoromount (Vector Labs), sections were analyzed using a TCS-Sp5X Confocal Fluorescence Microscope (Leica) at a resolution of 1024×1024 pixels/frame with an objective ×63 (0.5 μm image thickness).

**Analysis of polysomnographic data**. Quantification of vigilance states: W, SWS, and PS were scored by 5-s epochs based on the visual inspection of EEG/EMG signals. Hypnograms were then drawn using a custom script in Spike-2 (CED) to finally calculate standard parameters (mean ± SEM) for each vigilance state (quantities, percentage, number, and bout duration)[12].

EMG quantification: To objectively evaluate effects on muscle tone during SWS and PS in vGAT-shRNA (vs. Ctrl-shRNA) rats, we computed the nuchal EMG signals to extract muscle tone values for each SWS episode, PS episode and the last 25-s of the preceding SWS bout over the 12-h light period of day 30 post-surgery. Only consolidated PS episodes (longer than 45-s) were considered, first 10-s and last 5-s being eliminated to avoid transition states. A mean value of EMG during PS and SWS was obtained for each rat. Besides, we also compared the ratio EMG during the last 25-s of SWS preceding PS vs. EMG during PS in both groups[12], to eliminate potential posture changes of the animal between SWS and following PS that could induce biased EMG modifications. These ratios were represented in decibels.

Actimetry: It is obvious that the nuchal EMG does not give information on motor activities generated in the distal extremities where phasic twitches typically occur during natural PS (limbs, tail, ears, whiskers). Therefore, we did actimetric analyses of videos to quantify all body movements during PS episodes using MatLab routines we recently developed. The so-called actimetric value represents the number of pixels with modified gray pattern between two successive video frames (20 ms). The mean actimetric value was calculated for each SWS and PS episodes for each animal as the mean number of pixels modified per second during the 12 h considered. To further estimate amounts of motor events during PS in vGAT-shRNA vs. Ctrl-shRNA rats, we then defined in Ctrl rats the threshold as 99.5th percentile of mean actimetric value of consolidated PS bouts. We then applied this threshold to consolidated PS episodes of vGAT-shRNA rats to compute the number of motor events (defined as the pixel increase above the threshold and lasting >50ms) and the percentage of PS time that rats spend moving and twitching[12].

Spectral analysis: A standard fast Fourier transform analysis of parietal EEG was computed using a Spike-2 script in vGAT-shRNA and Ctrl-shRNA rats[12]. A mean spectrum was generated for each vigilance state with standard frequency ranges of EEG rhythms ($\delta$, 0.5–4.6 Hz; $\theta$, 5.1–8.9 Hz; $\sigma$, 9.9–14 Hz; $\beta$, 15–30 Hz).

Activity maps: To visually illustrate the continuous modification of EMG muscle tone and actimetry distribution during SWS and PS for each control and treated rats, we classified all SWS and PS episodes in function of their respective mean values (from higher to lower values). The gradual color scale (from 0, corresponding to blue, low EMG activity) to 1 (yellow, high EMG activity) was normalized to the 5th and 99.5th percentile of SWS values. The same scale was applied for SWS and PS. The x-axis of activity maps is the episodes duration normalized between 0 and $2\pi$. The actimetry motor events are represented by red dots as described before (see "Actimetry"). Finally, data extracted from activity maps were used to analyze in vGAT-shRNA rats the distribution of these actimetry motor events along the duration of each PS episode normalized by one-third duration. All SWS episodes, SWS episodes preceding PS, and PS episodes were computed for each animal of both groups.

**Quantitative analysis of double-labeled neurons**. We mapped singly labeled c-Fos+, FG+, and double-labeled c-Fos+/FG+ neurons in coronal brainstem sections taken at 150 μm intervals (from AP −10.1 to −13.3 mm to Bregma) of four PSR rats with a FG injection in lumbar cord. We also bilaterally mapped c-Fos+ and c-Fos+/GlyT2+ neurons in brainstem and lumbar sections (every 150 μm) of 19 rats (4 PSC, 5 PSD, 6 PSR, and 4 STEP). Sections were drawn and labeled cells plotted with an Axioskop microscope (Zeiss) equipped with a motorized X–Y-sensitive stage and a color video camera connected to a image analysis system (Mercator; ExploraNova). Cells of each type (single or double-labeled) were counted for each brainstem area considered and exported using Mercator

(ExploraNova). When a structure was present on several sections, the neurons counted were summed. For spinal sections, neurons were counted in Rexed's layers 10 sp containing motoneurons and 7–8 sp containing interneurons.

**Statistics**. Because of the modest number of animals in each experiment, non-parametric statistical tests were used. For the comparison of vigilance states and the number of labeled neurons across PSC, PSD, PSR and STEP conditions, Kruskal–Wallis tests were performed, followed by Mann–Whitney U tests to identify pairwise differences. The effect of PS recovery vs. baseline (for c-Fos/FG protocol) was analyzed with a Wilcoxon signed rank test. For the AAV results, statistical differences in state quantities, duration, number and mean duration of episodes, spectral analysis, EMG quantification and actimetry between Ctrl-shRNA and vGAT-shRNA groups were also determined with a Mann–Whitney U test. All statistics were performed using StatView software and a significant effect was considered when $p < 0.05$.

**Animal studies approval**. We state about our adherence to the ARRIVE guidelines (3Rs) relative to experimental research using animals. All experiments were conducted in accordance to the European Community Council Directive for the use of research animals (86/609/EEC, 2010/63/EU). Protocols and procedures used were approved by the local Ethical Committee (C2EA-55, Université Claude Bernard, Lyon I) and the Ministère de l'Enseignement Supérieur et de la Recherche (DR-2014–37).

**Data availability**. The authors declare that all relevant experimental data supporting the present study and illustrated in Figures and Supplementary Files are available from the corresponding author upon request.

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

## Acknowledgements

Warm thanks to Annabelle Bouchardon and Denis Ressnikoff from CIQLE facility for confocal microscopy, Yoan Chérasse for the AAV production and Anne-Laure Morel for ISH experiments, as well as Markus H. Schmidt for providing valuable edition of this manuscript. This work was supported by CNRS UMR5292, INSERM U1028, Université Claude Bernard Lyon I and the Agence Nationale de la Recherche (OPTOREM, ANR-13-BSV4-0003-01). S.V.G. received PhD grants from Ministère de l'Education Supérieure et de la Recherche and Association France Parkinson. P.F. was supported by grants from Association France Parkinson and Société Francaise de Recherche et Médecine du Sommeil (SFRMS).

## Author contributions

S.V.G., P.H.L., and P.F. designed the study. S.V.G., O.C., and P.F. collected and analyzed the anatomical data. S.V.G., F.B., P.A.L., and P.F. collected and analyzed the physiological, behavioral, and actimetry data. P.A.L. wrote Matlab routines for computing EEG, EMG and video raw data. SA helped for the surgery and animal caring. M.L. provided viral vectors. All authors discussed results. S.V.G., P.H.L., and P.F. designed the iconography and wrote the manuscript.

## Additional information

**Competing interests:** The authors declare no competing financial interests.

