## [Peer Review File · Nature Communications]

Reviewers' comments:

Reviewer #1 (Remarks to the Author):

In general, this is a well done study by a careful and productive group at Lyon. It is important because it shows dramatic evidence that medullary GABA/glycine neurons directly inhibit spinal motoneurons during the atonia of REM sleep.

I only have a few questions about this overall highly praiseworthy study.

The abolition of atonia was not complete, as illustrated by the need to have video evidence of distal muscle activity. It would've been helpful to have a more complete discussion of this absence of a total effect. One question, given the known often incomplete effect of shRNA of inactivating its target, was what was the level of GAD inactivation produced by this treatment? If incomplete, might explain some of the absence of the complete abolition of atonia. A related question would whether the degree of inhibition in each animal was associated with different levels of GAD inactivation. In any case the level of GAD reduction produced by shRNA treatment would be useful to know.

Another possibility of the lack of complete abolition of atonia might be lack of complete control by the ventral medulla neurons over REM sleep atonia.

Minor note. The mimicking of REM sleep behavior disorder is indeed clinically suggestive. However, no clear evidence of a synucleinopathy in the ventral medullary nuclei is presented. Of note, locus Coeruleus neurons are often affected by parkinsonism and other synucleinopathies, as cited in this paper.

Reviewer #2 (Remarks to the Author):

This is another ground-breaking study by the Lyon group utilizing modern, innovative experimental techniques that continues the longstanding Lyon tradition of exploring the mechanisms of experimentally-induced REM-without-atonias and REM sleep behavior disorder initiated by Michel Jouvet and colleagues in 1965. The results help shed additional light on the neuronal groups and pathways responsible for generating physiological REM-atonias in mammals.

A number of points need to be considered for minor revisions.

1) Title: given the reported findings, shouldn't a revised title be the following? "Genetic inactivation of ventromedial medulla inhibitory neurons induces REM sleep without atonia and REM sleep behavior disorder in rats"

2) Results, page 5: the entire page and the top 3 lines of page 6 comprise one paragraph, which is not good for the readers. New paragraphs could begin on line 123 and line 131.

3) Page 8: line 206 could begin a new paragraph.

4) Page 10: line 287 could begin a new paragraph.

5) Page 11: line 320 could begin a new paragraph.

6) The findings from this animal model should also be discussed (to the extent possible) within the context of what is found clinically in humans, in regards to the following scenarios:

i) REM-without-tonia (RWA) without clinical RBD (i.e. incidental polysomnographic finding in patients being evaluated for another complaint, such as snoring/sleep apnea).

ii) RWA with clinical RBD.

iii) Excessive phasic EMG activity in REM sleep, with preserved REM-atonia, in clinical RBD.

iv) RWA and excessive phasic EMG activity in REM sleep with clinical RBD.

v) Spontaneous vs. antidepressant (and other) medication-induced RBD for the above 4 scenarios.

The authors should cite and summarize the discussion about these experimental-clinical translational issues contained in the following publication:

Schenck CH, Mahowald MW. A novel animal model offers deeper insights into human REM sleep behaviour disorder. *Brain* 2017; 140 (2): 256-259.

7) Finally, at the end of Page 4, besides their comments of improving RBD treatments and patients' healthcare, the authors should also encourage future research on medication-induced RBD in pre-clinical animal models, to gain deeper insights into how disturbed neurochemical mechanisms induced pharmacologically by SSRIs, venlafaxine, TCAs, etc. can result in RWA, increased phasic EMG activity in REM sleep (with preserved REM-atonia: a very important consideration), and clinical RBD.

Reviewer #3 (Remarks to the Author):

The data in this manuscript examine the role of the VMM in controlling muscle atonia during PS sleep. The authors show that impairing GABA/glycine cells activity (using short hairpin RNA) in the VMM increases muscle tone during PS, and the authors conclude that these cells are responsible for causing muscle atonia in PS. The authors also indicate that removing GABA/glycine cell activity in the VMM causes violent movements in PS, and these movements may be akin to REM sleep behaviour disorder.

This is a straightforward, well-executed and generally well analyzed study that complements many previous studies showing that the VMM is a critical medullary region involved in regulating muscle activity during PS. However, the authors make a step forward by showing these cells are likely GABA/glycine containing.

The authors try to argue that there is a "controversy" concerning the location of the cells that mediate muscle atonia during PS. This argument seems contrived and unnecessary, and feels like a crafted attempt to make a "big deal" out of the obvious – i.e., more than one CNS region controls muscle tone in PS. It remains unclear why the authors argue this point, especially considering that several regions express cFOS after PS deprivation (ie., not just the VMM).

A major concern is that the authors show that muscle activity is increased in both SWS and PS (Fig

4F). This is a significant problem because it indicates that VMM “lesions” do not produce changes in muscle activity that are specific to PS. This finding indicated that the VMM more broadly controls muscle activity and that it is not just restricted to controlling activity in PS. Further, the author do not provide data to show if muscle activity is changed in waking behaviors, which is likely because cells in the VMM are associated with waking behaviors like eating (Weber et al. 2015).

The authors provide data showing that there is more movement in PS in animals with impaired GABA/glycine cell activity in the VMM, but do not demonstrate that these movements are restricted to PS. They need to show that there are no changes in movement in either SWS or waking in order to show that GABA/glycine cell activity is specific for PS and not more broadly associated with changing muscle activity in all behaviors.

Another major concern is that the authors do not impair VMM cell activity with any level of temporal specificity. If they are going to make the claim that the VMM is specifically associated with controlling muscle activity during PS they need to only impair GABA/glycine activity during that state or use methods that show some level of temporal specificity.

The authors also use cFos methods in an attempt to identify when VMM cells are active in relationship to PS. This method does not have adequate temporal specificity to resolve when these cells are active. Furthermore, PS deprivation also impairs the expression of SWS, making it impossible to resolve whether changes in cFos expression are associated with SWS or PS or both behaviors.

Overall, the author’s data do not directly support their narrow claims. The authors need to interpret their data and develop conclusions that are reflected by their experimental outcomes.

Reviewer #1 (Remarks to the Author)

In general, this is a well-done study by a careful and productive group at Lyon. It is important because it shows dramatic evidence that medullary GABA/glycine neurons directly inhibit spinal motoneurons during the atonia of REM sleep. I only have a few questions about this overall highly praiseworthy study.

We thank Reviewer 1 for the very positive comments.

Point 1: *The abolition of atonia was not complete, as illustrated by the need to have video evidence of distal muscle activity. It would've been helpful to have a more complete discussion of this absence of a total effect.*

- The actimetry based on video recordings is not able to measure directly putative loss of atonia during PS in vGAT-shRNA rats. We designed this method not because the abolition of atonia was not complete in EMG recordings, but to quantify abnormal movements occurring during PS in all body territories. It therefore brings a better and objective measurement of the rat's movements. As explained in the former manuscript, the nuchal EMG recording is not representative of the whole body, since the head does not move much during PS compared to distal territories even in Ctrl-shRNA rats. It is obvious that these abnormal movements indirectly suggest a major (even complete) loss of atonia in vGAT-shRNA animals since atonia is supposed to naturally block involuntary movements during PS.

- To conclude whether the abolition of atonia is complete as questioned by Referee 1 is a difficult task because it is indeed puzzling to objectively segregate tonic (muscle tone) and phasic (RBD) motor activities as illustrated on raw recordings from vGAT-shRNA and Ctrl-shRNA rats (Figure 4 C and D). These raw data are of particular importance since they allow to directly and clearly evidence the loss of atonia along the duration of PS episodes, including periods without RBD movements, although the muscle tonus is irregular with sudden and random variations (increased or decreased amplitude). Consequently, as shown in the new Figures 4F, the computerized data from nuchal EMG demonstrated a significant increase of muscle activity during PS compared to SWS in vGAT-shRNA rats and compared to PS in control animals. To confirm these data, the EMG SWS/PS ratio significantly decreased in Ctrl-shRNA rats and significantly increased in vGAT-shRNA ones (new Figures 4G). These data clearly show that the genetic inactivation of vGAT in ventromedial medulla (vmM) is followed by an overall increase of motor activity due to both loss of atonia and abnormal motor behaviors.

- To give a better and objective view of the dynamics of nuchal EMG and oneiric movements in PS and SWS we added the panels G and H in Figure 5, where values of nuchal EMG are pseudo-colored and phasic events are illustrated by red dots. They clearly show that there is a strong increase in muscle activity and movements in vGAT-shRNA rats compared to control animals.

Point 2: *One question, given the known often incomplete effect of shRNA of inactivating its target, was what was the level of GAD inactivation produced by this treatment? If incomplete, might explain some of the absence of the complete abolition of atonia. A related question would whether the degree of inhibition in each animal was associated with different levels of GAD inactivation. In any case the level of GAD reduction produced by shRNA treatment would be useful to know.*

- In this study, we choose to genetically inactivate mRNAs encoding the vGAT protein (and not GAD, the synthesizing enzyme for GABA), a vesicular protein essential for GABA/glycine neurotransmission, responsible both for GABA/glycine release in the synaptic cleft and reload in presynaptic vesicles. Targeting vGAT rather than GAD is a more efficient approach to fully block GABA/glycine neurotransmission from inhibitory neurons within vmM.

- In agreement with Referee 1 comment, the efficiency of shRNAs was a crucial point to address in priority. In that context, systematic post-mortem analysis of brains from Ctrl-shRNA and vGAT-shRNA rats was done by using *in-situ* hybridization of vGAT mRNAs to detect potential residual expression in AAV injection sites. As illustrated in Figure 3A and B, there was no residual expression of vGAT mRNAs within the injection sites of treated rats whereas such expression was not modified in injection sites of Ctrl-shRNA rats. In both cases, the injection sites delineated by epifluorescence emitted by reporter protein, mcherry were localized in the same structures and have the same extension (Figure 3A and B). Further, we confirmed by confocal microscopic analysis the loss of vGAT protein in axons emanating from inhibitory vmM neurons and projecting to lumbar motoneurons of vGAT-shRNA rats compared to control rats (Figure 3E).

- As also suggested by Referee 1, the extent of the genetic inactivation of vGAT mRNAs within the vmM was a parameter to be seriously considered at the end of the physiological experiments before interpretation. Based on the expression of fluorescent reporter protein mcherry and the extent of loss of vGAT mRNAs in vmM, only animals with a complete covering of the area containing neurons projecting to the spinal cord illustrated in Figure 2F were including in the physiological analysis. Rats with incomplete inactivation of vmM or misplacement of AAVs injection sites in surrounding vmM areas were not considered further. We therefore believe that GABA/glycine transmission was abolished in all vmM neurons projecting to the spinal cord. These consistent data demonstrate that both the expression of vGAT mRNAs and vGAT protein are completely suppressed in vmM GABA/glycine neurons in vGAT-shRNA rats.

Point 3: Another possibility of the lack of complete abolition of atonia might be lack of complete control by the ventral medulla neurons over REM sleep atonia.

As explained above, the genetic inactivation of the whole vmM is sufficient to induce powerful physiological effects on motor control during PS. That includes a major impact on muscle atonia with an increased tone in neck EMG during PS allowing the occurrence of intense abnormal motor behaviors mimicking RBD. In addition, by using c-Fos as a known marker of neuronal activity, we also report that inside the brainstem and the spinal cord, only the glyT2-positive (i.e. glycine in nature) neurons located in the vmM are recruited during PS hypersomnia. A very few were detected at spinal levels in line with previous intracellular recording in un-drugged cats showing that glycine Renshaw cells in lumbar cord are not activated during PS (Morales et al., 1988). Further, to eliminate the possibility of false negative results at spinal level, we designed an additional positive control by using treadmill to induce forced locomotion known to recruit inhibitory spinal neurons. As shown in Figure 2, glycine neurons in lumbar spinal cord are tremendously recruited during forced locomotion, but not during PS, confirming that the absence of c-Fos expression in these neurons during PS hypersomnia is not due to a technical problem. Taken together, our anatomical and functional data therefore clearly indicate that inhibitory neurons of vmM are responsible for the hyperpolarization of motoneurons during PS.

Point 4: Minor note. The mimicking of REM sleep behavior disorder is indeed clinically suggestive. However, no clear evidence of a synucleinopathy in the ventral medullary nuclei is presented. Of note, locus Coeruleus neurons are often affected by parkinsonism and other synucleinopathies, as cited in this paper.

We are in agreement with this remark of Referee 1. To date, there is no yet data in the literature reporting the presence of a synucleinopathic attack within vmM in RBD patients. There is just one case with a major inflammatory lesion of ventral medulla causing RBD (Limousin et al., 2009). In our paper, we hypothesized that an attack of inhibitory vmM neurons might be responsible for inducing RBD. Our proposal remains to be tested both in synuclein-injected animal and by anatomo-pathological analysis of RBD human tissues focused on vmM. We thus added two sentences in the Discussion to clarify this point (page 12, lines 347-353 and page 13, lines 363-364).

Reviewer #2 (Remarks to the Author)

This is another ground-breaking study by the Lyon group utilizing modern, innovative experimental techniques that continues the longstanding Lyon tradition of exploring the mechanisms of experimentally-induced REM-without-tonia and REM sleep behavior disorder initiated by Michel Jouvet and colleagues in 1965. The results help shed additional light on the neuronal groups and pathways responsible for generating physiological REM-atonia in mammals.

We are grateful to Referee 2 for these very positive comments.

A number of points need to be considered for minor revisions.

Point 1: *Title: given the reported findings, shouldn't a revised title be the following? "Genetic inactivation of ventromedial medulla inhibitory neurons induces REM sleep without atonia and REM sleep behavior disorder in rats"*

We agree that the proposed title of Reviewer 2 is more adapted to our study. Nevertheless, the guideline for Authors in Nature Communication stipulates that *"the title should be 15 words or fewer"* and the proposed title contains 19 words. We therefore propose to use the title "Genetic inactivation of ventromedial medulla inhibitory neurons induces REM sleep without atonia and REM sleep behavior disorders" of 17 words (Top page 1). We would like to ask for a special exemption to the Editor-in-Chief to validate our proposal.

Point 2: *Results, page 5: the entire page and the top 3 lines of page 6 comprise one paragraph, which is not good for the readers. New paragraphs could begin on line 123 and line 131.*

Done

Point 3: *Page 8: line 206 could begin a new paragraph.*

Done

Point 4: *Page 10: line 287 could begin a new paragraph.*

Done

Point 5: *Page 11: line 320 could begin a new paragraph.*

Done

Point 6: *The findings from this animal model should also be discussed (to the extent possible) within the context of what is found clinically in humans, in regards to the following scenarios:*

i) REM-without-atonia (RWA) without clinical RBD (i.e. incidental polysomnographic finding in patients being evaluated for another complaint, such as snoring/sleep apnea).

ii) RWA with clinical RBD.

iii) Excessive phasic EMG activity in REM sleep, with preserved REM-atonia, in clinical RBD.

iv) RWA and excessive phasic EMG activity in REM sleep with clinical RBD.

v) Spontaneous vs. antidepressant (and other) medication-induced RBD for the above 4 scenarios.

The authors should cite and summarize the discussion about these experimental-clinical translational issues contained in the following publication: Schenck CH, Mahowald MW. A novel animal model offers deeper insights into human REM sleep behaviour disorder. Brain 2017; 140 (2): 256-259.

This recently published paper has been added and cited in the discussion (page 12, line 345). We also added a new paragraph in the Discussion to put our data in rats in a more broadly clinical context (page 12, 343-53).

Point 7: *Finally, at the end of Page 4, besides their comments of improving RBD treatments and patients' healthcare, the authors should also encourage future research on medication-induced RBD in pre-clinical animal models, to gain deeper insights into how disturbed neurochemical mechanisms induced pharmacologically by SSRIs, venlafaxine, TCAs, etc. can result in RWA, increased phasic EMG activity in REM sleep (with preserved REM-atonia: a very important consideration), and clinical RBD.*

We would like to thank Reviewer 2 for this pertinent suggestion about a crucial point in clinical practice (side-effects of medications for depression and anxiety inducing RBD). It will be indeed important to address such question in the future in rats and mice to understand if those neuropharmacological treatments could induce a specific response in vmM neurons provoking this RBD side-effect. We completed the last sentence of the Introduction (page 4, lines 97-98) and in the Discussion (page 13, lines 366-369) to suggest that neurochemical mechanisms specifically targeting inhibitory vmM or excitatory SLD neurons may explain medication-induced RBD in patients with chronic antidepressant treatments and that animal pre-clinical models may help to test this hypothesis.

Reviewer #3 (Remarks to the Author):

Point 1: *The data in this manuscript examine the role of the VMM in controlling muscle atonia during PS sleep. The authors show that impairing GABA/glycine cells activity (using short hairpin RNA) in the VMM increases muscle tone during PS, and the authors conclude that these cells are responsible for causing muscle atonia in PS. The authors also indicate that removing GABA/glycine cell activity in the VMM causes violent movements in PS, and these movements may be akin to REM sleep behaviour disorder. This is a straightforward, well-executed and generally well analyzed study that complements many previous studies showing that the VMM is a critical medullary region involved in regulating muscle activity during PS. However, the authors make a step forward by showing these cells are likely GABA/glycine containing.*

We are very obliged to Reviewer 3 for acknowledging the breakthrough of the present work.

Point 2: *The authors try to argue that there is a “controversy” concerning the location of the cells that mediate muscle atonia during PS. This argument seems contrived and unnecessary, and feels like a crafted attempt to make a “big deal” out of the obvious – i.e., more than one CNS region controls muscle tone in PS. It remains unclear why the authors argue this point, especially considering that several regions express cFOS after PS deprivation (ie, not just the VMM).*

The word “controversy” was used once at the end of the Introduction to explain that there is still “a matter of debate” or “a true uncertainty” about the exact location within brainstem and/or spinal cord of GABA / glycine pre-motoneurons activated during PS and responsible for inducing muscle atonia during PS. Recent papers suggested that spinal inhibitory interneurons might play a crucial role (Krenzer et al., 2013; Weber et al., 2015; Vetrivelan et al., 2009), and this one of medullary reticular nuclei GABA/glycine neurons was not yet clearly defined . Therefore, there was a strong need to fully address and solve this question. To this aim, we first implemented a new experimental paradigm combining Fos immunodetection with *in-situ* hybridization of GlyT2 mRNAs, a specific maker of glycine neurons in PS hypersomniac rats to localize the glycine cell bodies activated during PS hypersomnia (not during PS deprivation). In line with Referee 3 comment, neurons labeled for c-Fos were indeed found in many brain areas from forebrain to lower brainstem including cortex as previously reported (for example, Verret et al, 2005, Sapin et al 2010, Renouard et al. 2015). However, our present double-labeling experiments revealed that neurons positive simultaneously for GlyT2 and c-Fos during PS hypersomnia were only found in the vmM. A weak number, if any were found in the lumbar spinal cord. Further, we found that vmM neurons directly project to lumbar motoneurons using anterograde tract-tracing with AAVs and that their selective inactivation induces PS without atonia and RBD. These original anatomical and functional data demonstrate for the first time that GABA/glycine vmM neurons inhibit motoneurons during PS.

Based on the comment of Referee 3, we thus replaced the word “controversy” by “To make a significant step forward in this debate” in the revised manuscript (page 3, line 87).

Point 3: *A major concern is that the authors show that muscle activity is increased in both SWS and PS (Fig 4F). This is a significant problem because it indicates that VMM “lesions” do not produce changes in muscle activity that are specific to PS. This finding indicated that the VMM more broadly controls muscle activity and that it is not just restricted to controlling activity in PS. Further, the author do not provide data to show if muscle activity is changed in waking behaviors, which is likely because cells in the VMM are associated with waking behaviors like eating (Weber et al. 2015).*

- We agree with Referee 3 that the former Figure 4F was not sufficiently explicit and might have induced a misinterpretation of our data. Indeed, visually, the mean value of EMG during SWS seems higher in vGAT-shRNA vs. Ctrl-shRNA rats (histogram bars). However, as indicated in the manuscript and legends of Figure 4F, there was no significant difference in EMG values during SWS between Ctrl-shRNA and vGAT-shRNA rats. Moreover, a careful observation of each individual rat (open and black circles) showed that values are exactly in the same range during SWS both for Ctrl-shRNA and vGAT-shRNA rats, excepting for outlier rats (one vGAT-shRNA and one Ctrl-shRNA) with higher values during SWS and PS, that likely biased the mean value of SWS EMG. To avoid misinterpretation of our data, we re-processed all EMG raw data for each Ctrl-shRNA and vGAT-shRNA rats, first by filtering 50hz frequencies (48Hz-52Hz band) that might increased individual values, and second by computing muscle activity during all SWS and PS episodes and not only during the 30-sec preceding PS occurrence as previously done. Please find enclosed, the mean spectral power of all animals before and after filtering artifactual frequencies during SWS and PS. Note that the EMG value during SWS is not different between groups. Only two outlier animals (Ctrl-shRNA 2 and vGAT-shRNA 1) are still slightly increased due to residual 50Hz frequencies. As it is maintained along the whole recording, we can compare SWS and PS within the same animal. As shown in the new Figure 4F, Ctrl-shRNA rats present the physiological diminution of muscle tone between SWS and

PS (so called muscle atonia), while EMG values are increased during PS in vGAT-shRNA rats.

- We modified Figure 4F using the complete and filtered data from the same initial samples of rats. We obtain the same results and significance than with the previous analysis confirming that the increased muscle activity in vGAT-shRNA rats is specific of PS. To illustrate the differences in the EMG between PS and SWS in both groups, we already showed the EMG ratio in Figure 4G. To illustrate more finely how nuchal EMG is modified between these two states, we plotted the same graphics in decibels.

- As already reported in the manuscript, and in contrast to Weber et al. (2015), we did not observe modification of eating and food intake in vGAT-shRNA rats and the treated animals increased their body weight along the experiments similarly to the control rats. As discussed in the manuscript (page 11, lines 307-315), Weber et al. (2015) injected their opto- and pharmacogenetic tools in the most lateral part of the ventral medulla including the LPGi and the lateral part of the vmM, avoiding its medial part. Their injection sites never involved the whole vmM. Inhibitory neurons within the LPGi send strong ascending projections, including to the locus coeruleus, known to be major wake-promoting, and the ventrolateral part of the periaqueductal grey matter, a PS permissive system (Weber et al., 2015; Rampon et al., 1999; Luppi et al., 1995, review Luppi et al., 2013). Therefore, the targeting of two different, partly overlapping contingents of inhibitory neurons likely explains the differences reported in food intake and PS in the two studies.

- Finally, concerning the motor activity during waking, we scrutinized closely the behaviors of our animals using video recordings. As reported in the text, we did not observed abnormal behaviors of treated animals during waking compared to control rats. Moreover, we reported that after 2-h of permanent waking during the treadmill experiments, the vmM contains a very few number of c-Fos expressing neurons and none were labeled for GlyT2 mRNAs indicating that GABA/glycine neurons of vmM are not recruited during waking and therefore do not play an essential role in waking and waking-related behavior. Finally, SWS motor activity was not modified in treated compared to Ctrl-shRNA (see above). Altogether, our results therefore clearly indicate that the effects obtained are specific to PS.

Animal	PS Mean Spectral Power (48-52 Hz)		SWS Mean Spectral Power (48-52 Hz)	
	Before 50 Hz filtering	After 50 Hz filtering	Before 50 Hz filtering	After 50 Hz filtering
CTRL-shRNA-1	0,9124	0,0133	0,8978	0,0139
CTRL-shRNA-2	53,5148	0,0538	56,2530	0,0525
CTRL-shRNA-3	0,7221	0,0106	1,2630	0,0208
CTRL-shRNA-4	0,7656	0,0102	0,8602	0,0121
CTRL-shRNA-5	0,6237	0,0048	0,7825	0,0084

Animal	SWS Mean Spectral Power (48-52 Hz)		SWS Mean Spectral Power (48-52 Hz)	
	Before 50 Hz filtering	After 50 Hz filtering	Before 50 Hz filtering	After 50 Hz filtering
vGAT-shRNA-1	678,6227	0,1718	671,0777	0,0747
vGAT-shRNA-2	4,4644	0,0910	2,9069	0,0553
vGAT-shRNA-3	96,3014	0,0876	93,0914	0,0051
vGAT-shRNA-4	6,7329	0,0870	0,1558	0,0059
vGAT-shRNA-5	1,2939	0,0237	1,0826	0,0207
vGAT-shRNA-6	5,9883	0,1194	0,4071	0,0062
vGAT-shRNA-7	4,9651	0,0885	2,0806	0,0438

Mean spectral power of 50Hz frequencies (band 48h-52Hz) during SWS and PS before and after filtering treatment of raw EMG data for each animal in both experimental groups.

Point 4: *The authors provide data showing that there is more movement in PS in animals with impaired GABA/glycine cell activity in the VMM, but do not demonstrate that these movements are restricted to PS. They need to show that there are no changes in movement in either SWS or waking in order to show that GABA/glycine cell activity is specific for PS and not more broadly associated with changing muscle activity in all behaviors.*

As suggested by Referee 3, we re-analyzed EMG values during SWS (see above). Further, to better illustrate and compare quantities of movements during SWS and PS, we add three new panels to Figure 5. Figure 5C represents the actimetry values during SWS and PS in both group of rats. It shows that motor activity during SWS is very low and comparable in Ctrl-shRNA and vGAT-shRNA rats, whereas during PS vGAT-shRNA rats show a strong increase of activity compared to Ctrl-shRNA. Figures 5G and 5H are time course maps of representative Ctrl-shRNA (5G) and vGAT-shRNA rats (5H), in which actimetry values and EMG values are illustrated across all SWS episodes, SWS episodes followed by a PS bouts and all PS episodes. By means of such illustration, it can clearly be seen that EMG values and motor

events measured by actimetry increase specifically during PS in vGAT-shRNA rats but not in control animals. For this specific point, we added in the revised manuscript a paragraph in the Materials and Methods section about the analysis process for the new activity maps (page 17, lines 492-500), a new paragraph in the Results section (page 8, lines 227-33) and we modified the caption of the new Figure 5 (page 25, lines 774-780).

Point 5: *Another major concern is that the authors do not impair VMM cell activity with any level of temporal specificity. If they are going to make the claim that the VMM is specifically associated with controlling muscle activity during PS they need to only impair GABA/glycine activity during that state or use methods that show some level of temporal specificity.*

In this comment, Referee 3 refers to other new methods like optogenetics indeed allowing reversible neuronal activation/inhibition with a very good temporal resolution. However, such approach is not easily available in rats, the species in which we harvested all experimental data on networks responsible for PS. Moreover, the irreversible genetic inactivation of the GABA/glycine neurons within vmM is to date the best way to obtain a model closely mimicking chronic and irreversible human RBD. Finally, the effects obtained are specific to PS with no statistical modification of the EMG during SWS and no effect on motor behavior during waking. In addition, although optogenetic experiments would be of interest, it is an invasive method needing to implant two 250 um-thick optic fibers bilaterally in a very tiny structure like the mouse vmM. In summary although of interest, we don't believe optogenetic would bring additional crucial information compared to our results.

Point 6: *The authors also use cFos methods in an attempt to identify when VMM cells are active in relationship to PS. This method does not have adequate temporal specificity to resolve when these cells are active. Furthermore, PS deprivation also impairs the expression of SWS, making it impossible to resolve whether changes in cFos expression are associated with SWS or PS or both behaviors.*

This technical point has been discussed many times in our previous c-Fos papers as well by other well-known sleep teams (Pr. CB Saper, BE Jones, M. Chase, etc) concluding that c-Fos is to date the best marker of neuronal activation in rodents. Besides, in a large number of papers including ours, every time c-Fos labeled neurons have been observed in a structure after PS hypersomnia, unit recordings confirmed the presence of neurons specifically active during PS (Goutagny et al., 2008, Boissard et al., 2002, Boucetta et al., 2014, etc). Further, without c-Fos, neurons critically involved in PS like the hypothalamic MCH ones would never have been discovered (Verret et al., 2003). Finally, PS-on neurons have been already recorded within the vmM of cats (Nakamura et al., 1986; Sakai et al., 1979) and rodents (Weber et al, 2005) in coherence with our c-Fos data. In none of these studies, SWS-active neurons were recorded in the structures containing c-Fos+ neurons. Further, SWS quantities are similar in control conditions and during PS hypersomnia (see Supplementary Table 1) and yet there is no cFos staining in control conditions suggesting that such quantity of SWS is not sufficient to induce cFos expression. In view of such numbers of congruent data, there is a minimal chance if any that the GABA/glycine vmM neurons expressing c-Fos during PS recovery are not neurons specifically activated during PS.

Point 7: *Overall, the author's data do not directly support their narrow claims. The authors need to interpret their data and develop conclusions that are reflected by their experimental outcomes.*

This concluding comment of Reviewer 3 is quite surprising, further when comparing with the positive introductory comment (see Point 1). We consider that the present paper provides congruent data achieved with different experimental approaches and of sufficient strength to support our conclusion. We hope that our answers to requests and comments will be satisfactory. As suggested by Referee 3, we dampen our claims with a less assertive and more cautious style of writing along the whole revised manuscript.

Reviewers' comments:

Reviewer #2 (Remarks to the Author):

There is one more part that needs revision, related to Reviewer #1, Point 4:

The authors failed to cite a pertinent paper on medullary alpha-synuclein neuropathology in RBD. So for revision the authors should state the following: There is in fact a reported case of synuclein attack in the vmM of a 72 year old male idiopathic RBD patient with a 15 year history of dream-enactment, who died of pneumonia. [Citation #1 below] Parkinsonism was never detected nor dementia during serial neurological examinations within this 15 year period. Neuropathological examination revealed Lewy Body Disease. As shown in Table 1 of that report, which lists the distribution of alpha-synuclein pathology and neuronal loss, the Medullary RF had 2+ (moderate) Lewy bodies, 2+ Lewy neurites, and 0 Neuronal loss. Quantifying neuronal loss is far more important than identifying alpha-synucleinopathy pathology, according to Boeve BF, since the presence of a protein deposit does not necessarily mean that this is the cause of clinical symptoms [Citation #2]

1. Boeve BF, Dickson DW, Olson EJ, et al. Insights into REM sleep behavior disorder pathophysiology in brainstem-predominant Lewy body disease. *Sleep Medicine* 2007; 8: 60-64.

2. Boeve BF, Silber MH, Ferman TJ, et al. Clinicopathologic correlations in 172 cases of rapid eye movement sleep behavior disorder with or without a coexisting neurologic disorder. *Sleep Med* 2013; 14: 754-62.

Reviewer #3 (Remarks to the Author):

The authors have responded nicely to my previous comments and concerns. In general, the paper is improved, but there are still many remaining grammar mistakes and awkward language that needs to be corrected. For example, some of the section headings in the results are oddly worded (e.g. "Induced oneiric-like motor behaviors during PS without atonia" and "Where are localized the PS-on neurons potentially responsible for the glycine-mediated hyperpolarization of somatic motoneurons during PS?").

The authors have more convincingly shown that the effect of "lesioning" vm GABA cells is primarily restricted to PS as opposed to SWS (Figs 4 and 5). This update and clarification alleviates my major concern about the results and their interpretation. A very nice update to the paper and data!

I only have a few additional comments and suggestions at this point.

1) The authors indicate that percentage of PS bouts with increased muscle activity are higher in the vm lesioned animals versus the controls. I think it would be helpful to clearly indicate "how many" of the total number of PS bouts exhibit elevated muscle activity in the lesioned animals. This needs to be done. And this is important in the context of RBD since not all PS episodes in human RBD patients exhibit elevated muscle activity.

2) In Fig 5 the authors indicate that oneiric behaviors occur during PS, but they do not indicate how frequently these behaviors occur. I suggest that the authors quantify how many times these behaviors occur. Do they see these behaviors in every PS episode or are they infrequent? This needs to be quantified if possible.

3) The authors refer to PS atonia, but do not actually measure atonia, instead they measure overall muscle activity across PS bouts. In RBD patients there is a loss of PS atonia and an increase in the number of phasic muscle twitches. Currently, the authors do not strictly differentiate this in their EMG quantification of nuchal activity and they may want to indicate this in their manuscript. Brooks and Peever (Current Biology, 2016) have developed methods to objectively quantify “basal muscle tone” (atonia) and “muscle twitches” during PS in rats, and they show that there’s a temporal distribution of both aspects of muscle activity across PS episodes. I’m not suggesting that the authors necessarily need to quantify these variables, but it would be interesting and helpful to at least comment on whether there is a temporal distribution of motor activity across individual PS episodes. For example, is there more motor activity towards the end of each PS episode and least at the beginning of each episode as suggested by Brooks/Peever Current Biology 2016?

Editorial Concerns:

In addition to the Reviewer comments above please also address these following concerns:

- 1) The authors indicate that they made AAV injections to delete VGAT in the vmM and that these injections prevented GABA/glycine cell function in the vmM, leading to a partial removal of REM atonia. However, the authors also indicate that they made injections that depleted GABA/glycine cell activity in areas “outside” the vmM. It would be useful to present these data to show how such “off-target” manipulations influenced REM atonia.
- 2) The authors show that removing GABA/glycine cells function in the vmM partial blocks REM atonia, and they make this claim by showing the muscle activity increase in REM sleep to SWS levels. However, the authors never show/compare how vmM inactivation influences muscle activity “relative to wakefulness”. It would be very important to show if vmM inactivation elevates levels muscle activity to normal “waking levels”.
- 3) The authors indicate that there is no current literature showing synucleinopathy within the vmM. This is incorrect. The Lancet Neurology paper by Iranzo et al. (2013) clearly shows evidence of Lewy pathology in the vmM region in patients with Parkinson’s disease and RBD. This should be mentioned in a revised manuscript.

Reviewer #1 (Remarks to the Author)

No comment received.

Reviewer #2 (Remarks to the Author)

Point 1: *There is one more part that needs revision, related to Reviewer #1, Point 4*

The authors failed to cite a pertinent paper on medullary alpha-synuclein neuropathology in RBD. So for revision the authors should state the following: There is in fact a reported case of synuclein attack in the vmM of a 72 year old male idiopathic RBD patient with a 15 year history of dream-enactment, who died of pneumonia. [Citation #1 below] Parkinsonism was never detected nor dementia during serial neurological examinations within this 15 year period. Neuropathological examination revealed Lewy Body Disease. As shown in Table 1 of that report, which lists the distribution of alpha-synuclein pathology and neuronal loss, the Medullary RF had 2+ (moderate) Lewy bodies, 2+ Lewy neurites, and 0 Neuronal loss. Quantifying neuronal loss is far more important than identifying alpha-synucleinopathy pathology, according to Boeve BF, since the presence of a protein deposit does not necessarily mean that this is the cause of clinical symptoms [Citation #2].

1. Boeve BF, Dickson DW, Olson EJ, et al. Insights into REM sleep behavior disorder pathophysiology in brainstem-predominant Lewy body disease. Sleep Medicine 2007; 8: 60-64.

2. Boeve BF, Silber MH, Ferman TJ, et al. Clinicopathologic correlations in 172 cases of rapid eye movement sleep behavior disorder with or without a coexisting neurologic disorder. Sleep Med 2013;14:754-62.

We thank Reviewer 2 for pointing out these two studies. Indeed, they strengthen the hypothesis of a synucleinopathic attack of the vmM in patients with RBD and Parkinson's disease. The first paper was cited in the former version of the manuscript (page 12, line 350). As suggested, we further added a paragraph discussing this point (Discussion, Pages 12-13, Lines 368-378).

Reviewer #3 (Remarks to the Author):

The authors have responded nicely to my previous comments and concerns. In general, the paper is improved, but there are still many remaining grammar mistakes and awkward language that needs to be corrected. For example, some of the section headings in the results are oddly worded (e.g. "Induced oneiric-like motor behaviors during PS without atonia" and "Where are localized the PS-on neurons potentially responsible for the glycine-mediated hyperpolarization of somatic motoneurons during PS?").

The authors have more convincingly shown that the effect of "lesioning" vM GABA cells is primarily restricted to PS as opposed to SWS (Figs 4 and 5). This update and clarification alleviates my major concern about the results and their interpretation. A very nice update to the paper and data!

We wish to warmly thank Reviewer 3 for these positive comments.

I only have a few additional comments and suggestions at this point.

Point 1: The authors indicate that percentage of PS bouts with increased muscle activity are higher in the vM lesioned animals versus the controls. I think it would be helpful to clearly indicate "how many" of the total number of PS bouts exhibit elevated muscle activity in the lesioned animals. This needs to be done. And this is important in the context of RBD since not all PS episodes in human RBD patients exhibit elevated muscle activity.

We fully agree with Referee 3. Thus, we calculated and now report in the manuscript the percentage of PS episodes with an elevated muscle activity in vGAT-shRNA rats (Results, Page 8, Lines 212-213).

Point 2: In Fig 5 the authors indicate that oneiric behaviors occur during PS, but they do not indicate how frequently these behaviors occur. I suggest that the authors quantify how many times these behaviors occur. Do they see these behaviors in every PS episode or are they infrequent? This needs to be quantified if possible.

The Figure 5D previously showed that the mean number of motor events was increased significantly during PS in vGAT-shRNA rats. As suggested by Referee 3, we extracted the data from the activity maps to calculate the mean percentage of PS episodes with motor events. Based on such analysis, we now show that the abnormal motor events occur in more than 70% of the PS episodes in treated rats (Results, Page 8, Lines 226-227).

Point 3: The authors refer to PS atonia, but do not actually measure atonia, instead they measure overall muscle activity across PS bouts. In RBD patients there is a loss of PS atonia and an increase in the number of phasic muscle twitches. Currently, the authors do not strictly differentiate this in their EMG quantification of nuchal activity and they may want to indicate this in their manuscript. Brooks and Peever (Current Biology, 2016) have

developed methods to objectively quantify “basal muscle tone” (atonia) and “muscle twitches” during PS in rats, and they show that there’s a temporal distribution of both aspects of muscle activity across PS episodes. I’m not suggesting that the authors necessarily need to quantify these variables, but it would be interesting and helpful to at least comment on whether there is a temporal distribution of motor activity across individual PS episodes. For example, is there more motor activity towards the end of each PS episode and least at the beginning of each episode assented by Brooks/Peever *Current Biology* 2016?

The activity maps (with two examples added to Figure 5 in response to the previous comments of Reviewer 3) were provided to show the temporal distribution of muscle tone changes and abnormal movements along individual PS episodes. Based on these maps, it is clear that abnormal movements occur preferentially during the second half of PS episodes. However, and as suggested by Referee 3, an objective quantification was necessary to demonstrate it. Thus, we calculated the mean percentage of motor events occurring during the first, second or last third of PS episodes. We now show that half of these motor events are generated during the last third of PS episodes, in agreement with Brooks and Peever (2016, Results, Page 9, Line 244). We completed these analyses by calculating the percentage of time spent moving in the first, second and last third of PS bouts. The new data have been added to the revised manuscript (Results, Page 9, Lines 241-247).

Editorial Concerns:

In addition to the Reviewer comments above please also address these following concerns:

Point 1: The authors indicate that they made AAV injections to delete VGAT in the vmM and that these injections prevented GABA/glycine cell function in the vmM, leading to a partial removal of REM atonia. However, the authors also indicate that they made injections that depleted GABA/glycine cell activity in areas “outside” the vmM. It would be useful to present these data to show how such “off-target” manipulations influenced REM atonia. In the former version of the manuscript, we wrote “Animals with incomplete vmM transfection or with a spread of AAV encroaching laterally the LPGi and dorsally the Gi were not considered further”. We have no animal specifically injected in medullary areas surrounding the vmM, since as showed by the present anatomical study, neither LPGi nor Gi (nor any other brainstem areas) contain GABA/glycine inhibitory neurons specifically activated during PS AND projecting to spinal lumbar cord. Our control experiments are rats with injection of Ctrl-shRNA in vmM inducing no depletion of GABA/glycine in vmM neurons.

To avoid other potential confusion or misinterpretation, we modified the early sentence in the revised version (Results, Page 7, Lines 190-195).

Point 2: The authors show that removing GABA/glycine cells function in the vmM partial blocks REM atonia, and they make this claim by showing the muscle activity increase in REM sleep to SWS levels. However, the authors never show/compare how vmM inactivation influences muscle activity “relative to wakefulness”. It would be very important to show if vmM inactivation elevates muscle activity to normal “waking levels”.

We would like to thank the Editor-in-chief for raising this point. During waking, rats display a variety of motor behaviors (walking, exploration of the home cage, eating, drinking, scratching, grooming... etc) with obvious strong impact on nuchal EMG recording. In the former version of the manuscript, we pointed out that no differences in motor activity during waking were observed between control and treated animals. To confirm our observations, we now provide a quantitative analysis. We calculated the EMG values during all waking episodes for each animal included. The new results show that the EMG values during waking are not significantly different between the two experimental groups (Page 8, Lines 215-218). It confirms that the effect reported is specific to PS. In addition, we included a video recording of a vGAT-shRNA rat displaying “normal” motor behaviors during active waking such as exploring and grooming (Results, Page 9, Line 265). However we consider this video without any necessary and crucial information for the readers.

Point 3: The authors indicate that there is no current literature showing synucleinopathy within the vmM. This is incorrect. The *Lancet Neurology* paper by Iranzo et al. (2013) clearly shows evidence of Lewy pathology in the vmM region in patients with Parkinson’s disease and RBD. This should be mentioned in a revised manuscript.

We are grateful to the Editor-in-Chief to point out the article of Iranzo et al. (2013). We now cite it in the text (Discussion, Page 12, Lines 366-368).